# ON THE EFFECT OF CONSENSUS IN DECENTRALIZED DEEP LEARNING

## ABSTRACT

Decentralized training of deep learning models enables on-device learning over networks, as well as efficient scaling to large compute clusters. Experiments in earlier works revealed that decentralized training often suffers from generalization issues: the performance of models trained in a decentralized fashion is in general worse than the performance of models trained in a centralized fashion, and this generalization gap is impacted by parameters such as network size, communication topology and data partitioning.

We identify the changing consensus distance between devices as a key parameter to explain the gap between centralized and decentralized training. We show that when the consensus distance does not grow too large, the performance of centralized training can be reached and sometimes surpassed. We highlight the intimate interplay between network topology and learning rate at the different training phases and discuss the implications for communication efficient training schemes. Our insights into the generalization gap in decentralized deep learning allow the principled design of better training schemes that mitigate these effects.

## 1 INTRODUCTION

Highly over-parametrized deep neural networks show impressive results in machine learning tasks, which also lead to a dramatic increase in the size, complexity, and computational power of the training systems. In response to these challenges, distributed training algorithms (i.e. data-parallel large mini-batch SGD) have been developed for use in data-center (Goyal et al., 2017; You et al., 2018; Shallue et al., 2018). These SOTA training systems in general use the All-Reduce communication primitive to perform exact averaging on the local mini-batch gradients computed on different subsets of the data, for the later synchronized model update. However, exact averaging with All-Reduce is sensitive to the communication hardware of the training systems, causing the bottleneck in efficient deep learning training. To this end, decentralized training has become an indispensable training paradigm for efficient large scale training in the data-center, alongside its orthogonal benefits on preserving users' privacy for edge AI (Bellet et al., 2018; Kairouz et al., 2019).

Several very recent papers aim to address the communication overhead in data-center training by compressing the gradients (Koloskova et al., 2020a; Vogels et al., 2020) or designing better a communication topology (Assran et al., 2019). However, decentralized training for deep learning models still often results in a severe loss in generalization performance, even after hyper-parameter fine-tuning (see our Table 1 as well as Tables 1, 2, 3 in Assran et al., 2019). This phenomenon is poorly understood even in relatively straightforward i.i.d. data distribution scenarios (i.e. the data-center case), to which very few works are dedicated (but none of them provide insights into the generalization gap).

| | Complete | Ring |
|---|---|---|
| n=16 | $92.91 \pm 0.12$ | $92.51 \pm 0.19$ |
| n=32 | $92.82 \pm 0.27$ | $91.74 \pm 0.15$ |
| n=64 | $92.71 \pm 0.11$ | $89.87 \pm 0.12$ |

Table 1: The generalization issues for decentralized deep learning on two topologies (ResNet-20 on CIFAR-10 with $n \in \{16, 32, 64\}$ workers). The test top-1 accuracies are over three seeds with fine-tuned learning rates.

In this work, we theoretically identify the consensus distance, i.e. the average discrepancy between the nodes, as the key parameter that captures the joint effect of decentralization. We show that there exists a critical consensus distance: when the consensus distance is lower than this critical value, the optimization is almost unhindered. With the insight derived from optimization convergence,

we further question the existence of the critical consensus distance for deep learning in terms of generalization, and identify the training phase when the critical consensus distance matters. We believe that the answers to these questions are valuable in practice, as they offer the possibility to design training strategies that strike appropriate trade-off between targeted generalization performance and affordable communication resources.

- We show that tracking the consensus distance over the training phases can explain the generalization gap between centralized and decentralized training.

- We show theoretically that when the consensus distance does not exceed a critical value, then decentralization exerts negligible impact to the optimization. We argue how this critical value depends crucially on the learning rate and the mixing matrix.

- Through the lens of consensus distance, we empirically investigate *what is the desirable level of consensus distance during different phases of training, in order to ensure high generalization performance.* Our extensive experiments on Computer Vision (CV) tasks (CIFAR-10 and ImageNet-32) as well as Natural Language Processing (NLP) tasks (transformer models for machine translation) do confirm that a critical consensus distance indeed exists, and that consensus distances higher than this critical value heavily penalize the final generalization performance. Surprisingly, we find that large consensus distance might be beneficial in later training phases after optimization has plateaued, leading to improved generalization, which is consistent with the observations in centralized Post-Local SGD (Lin et al., 2020b).

- Based on our findings, we propose practical guidelines on how to achieve favorable generalization performance with low communication expenses, on arbitrary communication networks.

While our numerical study mainly focuses on the data-center setting with homogeneous nodes, our findings also apply to decentralized training over time-varying topologies and the more difficult heterogeneous setting alike. Our findings not only explain previously proposed ad-hoc solutions (e.g. using All-Reduce in the first training phase as in Assran et al., 2019) but our insights will allow the principled design of better decentralized training systems in future work.

## 2 RELATED WORK

Gossip averaging (Kempe et al., 2003; Xiao & Boyd, 2004; Boyd et al., 2006) forms the backbone of many decentralized learning algorithms. The convergence rate of gossip averaging towards consensus among the nodes can be expressed in terms of the (expected) spectral gap of the mixing matrix. Lian et al. (2017) combine SGD with gossip averaging for deep learning and show that the leading term in the convergence rate $\mathcal{O}\left(\frac{1}{n\varepsilon^2}\right)$ is consistent with the convergence of the centralized mini-batch SGD (Dekel et al., 2012) and the spectral gap only affects the asymptotically smaller terms. Similar results have been observed very recently for related schemes (Scaman et al., 2017; 2018; Koloskova et al., 2019; 2020a;b). As the communication topology also impacts the cost per round (number of peer-to-peer communications), sparse topologies have been proposed and studied recently (Assran et al., 2019; Wang et al., 2019; Nadiradze et al., 2020). Whilst a few recent works focus on the impact of the topology on the optimization performance (Luo et al., 2019; Neglia et al., 2020), we here identify the consensus distance as a more canonical parameter that can characterize the overall effect of decentralized learning, beyond only the topology, through which we are able to provide deeper understanding on the more fine-grained impact of the evolution of the actual consensus distance, on the generalization performance of deep learning training.

Prior work identified the consensus distance as an important parameter that can affect optimization performance and convergence, and provide approaches to increase consensus: for instance Scaman et al. (2017); Sharma et al. (2019) perform multiple consensus steps per round, and Tsitsiklis (1984); Nedić & Ozdaglar (2009); Duchi et al. (2012); Yuan et al. (2016) choose carefully tuned learning rates. However, these work do not provide insights into how consensus distance at different training phases impacts the decentralized training, which is the main target of this work.

## 3 THEORETICAL UNDERSTANDING

In this section we consider decentralized training with stochastic gradient descent (D-SGD) without momentum, but we are using the momentum version in all our DL experiments.

### 3.1 NOTATION AND SETTING

The agents are tasked to solve a sum-structured optimization problem $f \colon \mathbb{R}^d \to \mathbb{R}$ of the form

$$f^\star := \min_{\mathbf{x} \in \mathbb{R}^d} \left[ f(\mathbf{x}) := \tfrac{1}{n} \sum_{i=1}^n f_i(\mathbf{x}) \right], \tag{1}$$

where the components $f_i \colon \mathbb{R}^d \to \mathbb{R}$ are distributed among the $n$ nodes and are given in stochastic form: $f_i(\mathbf{x}) := \mathbb{E}_{\xi \sim \mathcal{D}_i}[F_i(\mathbf{x}, \xi)]$, where $\mathcal{D}_i$ denotes the local data distribution on node $i \in [n]$. For data-center settings, where data is re-shuffled periodically among nodes, these distributions are identical, but in other scenarios there can be differences between nodes. In D-SGD, each agent $i \in [n]$ maintains local parameters $\mathbf{x}_i^{(t)} \in \mathbb{R}^d$, and updates them as:

$$\mathbf{x}_i^{(t+1)} = \sum_{j=1}^n w_{ij} \left( \mathbf{x}_j^{(t)} - \eta \nabla F_j(\mathbf{x}_j^{(t)}, \xi_j^{(t)}) \right), \tag{D-SGD}$$

that is, by a stochastic gradient step based on a sample $\xi_i^{(i)} \sim \mathcal{D}_i$, followed by gossip averaging with neighboring nodes in the network encoded by the mixing weights $w_{ij}$. As parameters can differ across nodes, we define $\bar{\mathbf{x}} := \tfrac{1}{n} \sum_{i=1}^n \mathbf{x}_i$ and $\mathbf{X} := [\mathbf{x}_1, \ldots, \mathbf{x}_n] \in \mathbb{R}^{d \times n}$, and $\bar{\mathbf{X}} := [\bar{\mathbf{x}}, \ldots, \bar{\mathbf{x}}] \equiv \mathbf{X} \tfrac{1}{n} \mathbf{1}\mathbf{1}^\top$.

**Assumption 1** (Mixing matrix). *Every sample of the (possibly randomized) mixing matrix* $\mathbf{W} = \{w_{ij}\} \in \mathbb{R}^{n \times n}$ *is doubly stochastic and there exists a parameter $p > 0$ such that*

$$\mathbb{E}_{\mathbf{W}} \left\| \mathbf{X}\mathbf{W} - \bar{\mathbf{X}} \right\|_F^2 \le (1-p) \left\| \mathbf{X} - \bar{\mathbf{X}} \right\|_F^2, \forall \mathbf{X} \in \mathbb{R}^{d \times n}. \tag{2}$$

This assumption covers a broad variety of settings (see e.g. Koloskova et al., 2020b), such as D-SGD with fixed (constant) mixing matrix with spectral gap $\rho$, with parameter $p = 1 - (1-\rho)^2 = \Theta(\rho)$, but also for randomly chosen mixing matrices, for instance random matchings.

**Assumption 2** (L-smoothness). *Each function $f_i(\mathbf{x}) \colon \mathbb{R}^d \to \mathbb{R}$, $i \in [n]$ is differentiable and there exists a constant $L \ge 0$ such that for each $\mathbf{x}, \mathbf{y} \in \mathbb{R}^d$: $\|\nabla f_i(\mathbf{y}) - \nabla f_i(\mathbf{x})\| \le L \|\mathbf{x} - \mathbf{y}\|$.*

**Assumption 3** (Bounded noise $\sigma$ and diversity $\zeta$). *There exists constants $\sigma^2, \zeta^2$ s.t. $\forall \mathbf{x}_1, \ldots \mathbf{x}_n \in \mathbb{R}^d$*

$$\tfrac{1}{n} \sum_{i=1}^n \mathbb{E}_{\xi_i} \|\nabla F_i(\mathbf{x}_i, \xi_i) - \nabla f_i(\mathbf{x}_i)\|_2^2 \le \sigma^2, \qquad \tfrac{1}{n} \sum_{i=1}^n \|\nabla f_i(\mathbf{x}_i) - \nabla f(\mathbf{x}_i)\|_2^2 \le \zeta^2. \tag{3}$$

### 3.2 DECENTRALIZED CONSENSUS OPTIMIZATION

Under the above assumptions, which are standard in decentralized optimization, the convergence rate of (D-SGD) has been shown to be:

**Theorem 1** (Koloskova et al. (2020b)). *Let $f_i$ be L-smooth and stepsize $\gamma \le \gamma_{\max} = \mathcal{O}\left(\tfrac{p}{L}\right)$. Then there exists an optimal stepsize $\gamma \le \gamma_{\max}$ such that $\tfrac{1}{T} \sum_{t=0}^{T-1} \mathbb{E} \left\| \nabla f(\bar{\mathbf{x}}^{(t)}) \right\|_2^2 \le \varepsilon$ for*

$$T = \mathcal{O}\left( \frac{\sigma^2}{n\varepsilon^2} + \frac{\sqrt{p}\sigma + \zeta}{p\varepsilon^{3/2}} + \frac{p}{\varepsilon} \right) \cdot L(f(\mathbf{x}_0) - f^\star).$$

In comparison, for centralized mini-batch SGD (C-SGD) we are allowed to choose a potentially much larger stepsize $\gamma \le \mathcal{O}\left(\tfrac{1}{L}\right)$, and can bound the number of iterations by $\mathcal{O}\left(\tfrac{\sigma^2}{n\varepsilon^2} + \tfrac{1}{\varepsilon}\right)$. While asymptotically both these rates are equivalent, they differ in low accuracy setting when $\varepsilon$ is not too small. That is, especially in the first phase of optimization where the lower order terms matter.

To measure differences between agents, we use the *consensus distance* $\Xi_t^2 := \tfrac{1}{n} \sum_{i=1}^n \left\| \bar{\mathbf{x}}^{(t)} - \mathbf{x}_i^{(t)} \right\|^2$.

**Remark 2** (Critical Consensus Distance (CCD)). *If the consensus distance is bounded by*

$$\Xi_t^2 \le \left( \frac{1}{Ln} \gamma \sigma^2 + \frac{1}{8L^2} \left\| \nabla f(\bar{\mathbf{x}}^{(t)}) \right\|^2 =: \Gamma_t^2 \right) \tag{4}$$

*for all t, then in D-SGD we may choose larger stepsizes $\gamma \le \gamma'_{\max} = \mathcal{O}\left(\tfrac{1}{L}\right)$ and recover the convergence rate of C-SGD, that is $\mathcal{O}\left(\tfrac{\sigma^2}{n\varepsilon^2} + \tfrac{1}{\varepsilon}\right)$ (Dekel et al., 2012; Bottou et al., 2018). We denote $\Gamma_t^2$ as* critical consensus distance *(CCD).*

The proof can be found in the Appendix A.1. Note that the CCD does not depend on the graph topology and that $\Gamma_t^2 > 0$, which means that we do not need perfect consensus between agents to recover the C-SGD rate, but we allow consensus distance $\Xi_t^2 \ge 0$ (i.e. the $\Xi_t^2 = 0 \ \forall t$, as we have for centralized optimization is sufficient, but not necessary).

We now estimate the magnitude of the consensus distance in D-SGD and compare it to CCD.

**Proposition 3** (Typical consensus distance). *Let $\phi_t^2 := \frac{1}{n}\sum_{i=1}^n \left\|\nabla f_i(\mathbf{x}_i^{(t)})\right\|^2$. Then under the assumption that $\gamma, p$ are constant, and the $\phi_t$ do not change too fast between iterations, i.e. not decreasing faster than exponentially: $\phi_t^2 \le (1+p/4)\phi_{t+1}^2$, the consensus distance in D-SGD satisfies*

$$\Xi_t^2 = (1-p)\gamma^2 \cdot \mathcal{O}\left(\frac{\phi_t^2}{p^2} + \frac{\sigma^2}{p}\right) . \tag{5}$$

We give the proof in the Appendix A.2. While these assumptions do not hold in epochs with learning rate decay, we observed in practice that when the learning rate is constant indeed the gradients do not change too fast and found it to be a reasonable approximation to capture the practical behavior (see Figure 5(b)).

### 3.3 CONTROLLING THE CONSENSUS DISTANCE

In this section we investigate scenarios where the typical consensus distance derived in Proposition 3 *can* be smaller than its critical value (CCD). This reveals two orthogonal strategies to control the consensus distance in D-SGD. We here assume diversity $\zeta = 0$ as with iid training data, and that the stepsize $\gamma \le \mathcal{O}\left(\frac{1}{L}\right)$ as for C-SGD, and give a more refined discussion in the appendix.

**Learning rate decay (changing $\gamma$).** We observe that when $\gamma = \mathcal{O}\left(\frac{p}{nL}\right)$ then $\Xi_t^2 \le \Gamma_t^2$ (if the noise $\sigma$ is small, especially for $\sigma = 0$, then the weaker assumption $\gamma = \mathcal{O}\left(\frac{p}{L}\right)$ is sufficient). However, choosing the stepsize too small can impact performance in practice. In C-SGD the constraint on the stepsize is loose ($\gamma \le \frac{1}{L}$). Yet, our observations show that after sufficient learning rate decay, the desired CCD can be reached.

**More gossip iterations (changing $p$).** We observe that when $\frac{1}{1-p} = \mathcal{O}(1 + \gamma Ln)$, then $\Xi_t^2 \le \Gamma_t^2$ (again, when the noise $\sigma$ is small, especially when $\sigma^2 = 0$, a weaker condition $\frac{1}{1-p} = \mathcal{O}(1 + \gamma L)$ is sufficient). Whilst designing new mixing topologies to control $p$ might not be possible due to practical constraints (fixed network, denser graphs increase latency, etc.) a simple and commonly used strategy is to use repeated gossip steps in every round.

**Lemma 4** (Repeated gossip). *Suppose $\mathbf{W} = \mathbf{W}_k \ldots \mathbf{W}_1$, for $k$ (possibly randomized) mixing matrices with parameter $p$ each. Then the mixing parameter for $\mathbf{W}$ is at least $p_{\mathbf{W}} \ge 1 - (1-p)^k$.*

From this, we see that the mixing parameter can be improved exponentially when applying more gossip steps. To ensure $p_{\mathbf{W}} \ge 1 - \frac{1}{1+\gamma Ln}$, at most $k \le \frac{\ln(1+\gamma Ln)}{p} = \tilde{\mathcal{O}}\left(\frac{1}{p}\right)$ repetitions are required.

These arguments show, that we can—at least in theory—recover the convergence behavior of C-SGD by controlling the consensus distance. We will now present numerical evidence, that corroborates these findings for deep learning tasks.

## 4 INSPECTING CONSENSUS DISTANCE FOR DECENTRALIZED TRAINING

Motivated by theoretical analysis in Section 3, we empirically investigate to *what extent our theory holds* and *how consensus distance interacts with deep learning training*. First we introduce and justify our experimental design in Section 4.1. We describe the implementation in Section 4.2. In Section 4.3, we present our findings on image classification benchmark with standard SGD optimizer, which is the main focus of this work; a preliminary study on Transformer with Adam optimizer and inverse square root learning rate schedule can be found in Section 4.4.

### 4.1 EXPERIMENT DESIGN: CONTROLLED DECENTRALIZED TRAINING PHASES.

**Phase-wise training.** As consensus distance evolves throughout training, identifying its impact at every training step is infeasible. Also, since consensus distance and critical consensus distance (CCD) have different but significant dependencies on learning rate (Remark 2 and Proposition 3), we would expect observations to be distinct over different learning rate phases but rather consistent within each phase. On CV tasks, stage-wise learning rate schedule is the common practice for SOTA training as described in Section 4.2: thus the training can be naturally divided into phases with the corresponding learning rate, in each of which key training dynamics are significantly different from the others, such as $\Xi_t$ (Figure 1), $\phi_t$ (Figure 5(b)) and $L$-smoothness (Figure 5(c)). The transformer

(NLP task) has no well-defined training phases due to the conventional inverse square root learning rate, thus for the sake of simplicity, we consider the entire transformer training as one phase as a preliminary study.

**Individual phase investigation.** In order to eliminate the coupling of effects from other phases, in each experiment we place only one phase under consensus distance control, while performing exact averaging (All-Reduce)[1] on model parameters for the other unstudied phases. For the ease of presentation, the term "phase-$x$" refers to a training phase between $x-1$-th and $x$-th learning rate decay. The notation "dec-phase-$x$" indicates that only in "phase-$x$" the model is trained with a decentralized communication topology, while for other phases we perform All-Reduce on model parameters. We compare the result of individual decentralized investigation with that of All-Reduce centralized training (on all training phases), so as to identify when (which phase) and how decentralized training influences final generalization performance[2].

## 4.2 EXPERIMENTAL SETUP

**Datasets and models.** We empirically study the decentralized training behavior on the following two tasks, on CNN and transformer architectures: (1) Image Classification for CIFAR-10 (Krizhevsky & Hinton, 2009) and ImageNet-32 (i.e. image resolution of 32) (Chrabaszcz et al., 2017), with the standard data augmentation and preprocessing scheme (He et al., 2016); and (2) Neural Machine Translation for the Multi30k dataset (Elliott et al., 2016). For Image Classification, ResNet-20 (He et al., 2016) with different widths are used on CIFAR (default width of 1) and ImageNet-32 (width factor of 3). For Neural Machine Translation, a down-scaled transformer architecture (by $2\times$ w.r.t. the base model in Vaswani et al. (2017)) is used. Weight initialization schemes follow Goyal et al. (2017); He et al. (2015) and Vaswani et al. (2017) respectively. Unless mentioned otherwise, our experiments are repeated over three random seeds.

**Training schemes.** We use mini-batch SGD with a Nesterov momentum of 0.9 without dampening for image classification task, and Adam for neural machine translation task. Unless mentioned otherwise we use the optimal learning rate (lr) for centralized training[3] in order to observe the impact of *decentralization* on normal *centralized* training.

- For image classification experiments, unless mentioned otherwise the models are trained for 300 and 90 epochs for CIFAR-10 and ImageNet-32 respectively; the local mini-batch size are set to 32 and 64. By default, all experiments use learning rate scaling and warmup scheme (Goyal et al., 2017). The learning rate is always gradually warmed up from a relatively small value (i.e. 0.1) for the first 5 epochs. Besides, the learning rate will be divided by 10 when the model has accessed specified fractions of the total number of training samples (He et al., 2016); we use $\{\frac{1}{2}, \frac{3}{4}\}$ and $\{\frac{1}{3}, \frac{2}{3}, \frac{8}{9}\}$ for CIFAR and ImageNet respectively. All results in tables are test top-1 accuracy.
- For experiments on neural machine translation, we use standard inverse square root learning rate schedule (Vaswani et al., 2017) with local mini-batch size 64. The warmup step is set to 4000 for the mini-batch size of 64 and is linearly scaled down by the global mini-batch size.

**Consensus distance control.** For consensus control, we adopt the "more gossip iterations" strategy introduced in Section 3.3. That is, we perform multiple gossip steps (if needed) until reaching the desired target consensus distance value. Two metrics are considered to set the consensus distance target value during the specified training phase:

- constant consensus distance (main approach[4]): the target consensus distance $\Xi$ for a phase is the *maximum consensus distance* $\Xi_{\max}$ of the *current phase* in normal (uncontrolled) decentralized training, multiplied by a factor. For a given topology, the smaller the factor, the tighter consensus.

---

[1] All-Reduce is used for all nodes to reach an exact consensus on model parameters.

[2] We demonstrate in Table 4 of Section 4.3 that the decentralization impacts on different phases are rather orthogonal, which justifies our design of examining one phase at a time.

[3] We do tune the learning rate, but its choice does not change the conclusion. E.g., no significant difference can be found for the curves at phase-1 for "ring (fine-tuned lr)" and "dec-phase-1 ($\Xi_{\max}$)" in Figure 2(a) and 2(b). We have similar observations in Table 11 after the sufficient learning rate tuning on phase-1.

[4] We use this one primarily since we can regulate the consensus distance in absolute values. We abuse the notation a bit by using $\Xi_{\max}$ alone to indicate the decentralized training w/o consensus distance control.

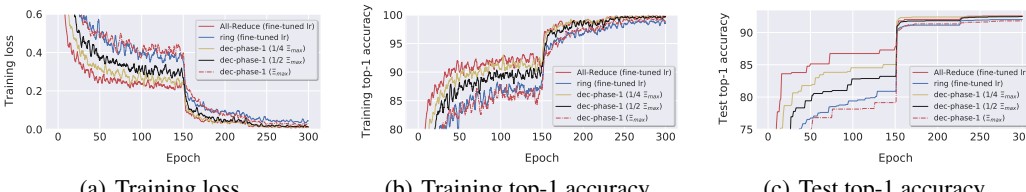

| (a) Training loss. | (b) Training top-1 accuracy. | (c) Test top-1 accuracy. |

Figure 2: Learning curves for ResNet-20 on CIFAR-10 ($n\!=\!32$). We compare tuned decentralized training (i.e. "ring") with dec-phase-1 on different target consensus distances.

Table 2: **The impact of consensus distance of different phases on generalization performance** (test top-1 accuracy) of training ResNet-20 on CIFAR-10. The All-Reduce performance for $n = 32$ and $n = 64$ are $92.82 \pm 0.27$ and $92.71 \pm 0.11$ respectively. The tuned decentralized performance (all phases on a fixed ring and w/o consensus distance control) for $n\!=\!32$ and $n\!=\!64$ are $91.74 \pm 0.15$ and $89.87 \pm 0.12$ respectively.

| target $\Xi$ 
 # nodes | dec-phase-1 | | | dec-phase-2 | | | dec-phase-3 | | |
|---|---|---|---|---|---|---|---|---|---|
| | $\Xi_{max}$ | $1/2\,\Xi_{max}$ | $1/4\,\Xi_{max}$ | $\Xi_{max}$ | $1/2\,\Xi_{max}$ | $1/4\,\Xi_{max}$ | $\Xi_{max}$ | $1/2\,\Xi_{max}$ | $1/4\,\Xi_{max}$ |
| n=32 | $91.78 \pm 0.35$ | $92.36 \pm 0.21$ | $92.74 \pm 0.10$ | $93.04 \pm 0.01$ | $92.99 \pm 0.30$ | $92.87 \pm 0.11$ | $92.60 \pm 0.00$ | $92.82 \pm 0.21$ | $92.85 \pm 0.24$ |
| n=64 | $90.31 \pm 0.12$ | $92.18 \pm 0.07$ | $92.45 \pm 0.17$ | $93.14 \pm 0.04$ | $92.94 \pm 0.10$ | $92.79 \pm 0.07$ | $92.23 \pm 0.12$ | $92.50 \pm 0.09$ | $92.60 \pm 0.10$ |

- adaptive consensus distance (in Appendix C.3.1): the target consensus distance $\Xi$ for the current step is the averaged local gradient norm $\phi_t^{avg}$ scaled by a factor. For stability, we use the exponentially moving averaged value $\phi_t^{ema}$ of $\phi_t^{avg}$ (accumulated during the corresponding phase).

We use a ring as our (main) decentralized communication topology[5] as it is a particular hard instance with large spectral gap (cf. Table 7), allowing to study a wide range of target consensus distances by modifying the number of communication rounds.

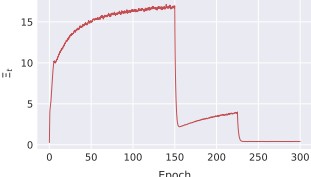

Figure 1: **The consensus distance** $\Xi$ for ResNet-20 on CIFAR-10 ($n\!=\!32$) with ring.

### 4.3 FINDINGS ON COMPUTER VISION TASKS

In this section we present our empirical findings and provide insights into how consensus distance at different phases impacts the training generalization for CV tasks (i.e. CIFAR-10, Imagenet-32).

**Critical consensus distance exists in the initial training phase and ensures good optimization and generalization.** In the initial training phase, both training and generalization performance are heavily influenced by the consensus distance[6] ("dec-phase-1" in Figure 2 and Table 2). A smaller consensus distance in the early phase results in considerably faster optimization (training loss) and higher generalization performance (test accuracy) and these advantages persist over the entire training.

When the consensus is more loose (e.g. $1/2\,\Xi_{max}$ for CIFAR-10), although the optimization (training performance) can eventually catch up with the centralized convergence (c.f. Figure 2(a) and 2(b)), a considerable generalization gap still remains (92.36 v.s. 92.82 for the setup in Figure 2) as shown in Table 2. This pattern is consistent[7] in ImageNet-32 experiments, as shown in Table 3. These observations to some extent are also consistent with the insights of the critical learning phase described in Jastrzebski et al. (2020); Frankle et al. (2020); Golatkar et al. (2019) for traditional (centralized) training, where it is argued that the initial learning phase is crucial for the final generalization.

Notably, perfect consensus distance is not required to recover the centralized training performance. For instance, $1/4\,\Xi_{max}$ is sufficient in CIFAR-10 experiments to approach the optimal centralized training performance in both optimization and *generalization* at the end. Smaller distances (e.g. $1/8\,\Xi_{max}$, $1/16\,\Xi_{max}$) do not bring significant gain (92.77 and 92.72 respectively in Table 9). The performance saturates (c.f. 92.74 for $1/4\,\Xi_{max}$) with significantly increased communication overhead (e.g. Figure 9 of Appendix C.1). This confirms that our analysis and discovery in Section 3 are

---

[5] We also study the effect of using different communication topologies as the base communication topology in Table 17 and 16. The observations are consistent with those of our main experiments with a ring topology.

[6] Equivalent results for SGD without momentum can be found in Table 12 in the appendix.

[7] In the case of ImageNet-32 with $n\!=\!32$, $1/2\,\Xi_{max}$ has already been tight enough to recover the centralized training performance. But a significant performance drop can be observed between $\Xi_{max}$ and $1/2\,\Xi_{max}$.

Table 3: **The impact of different consensus distances on generalization for different phases** of training ResNet-20-3 on ImageNet-32. The centralized baseline performances for $n=16$ and $n=32$ are $51.74 \pm 0.06$ and $51.98 \pm 0.37$ respectively, while those of decentralized training (on a fixed ring) are $51.04 \pm 0.06$ and $50.17 \pm 0.04$. The reported test top-1 accuracies are over two seeds.

| target Ξ  # nodes | dec-phase-1 | | | dec-phase-2 | | | dec-phase-3 | | | dec-phase-4 | | |
|---|---|---|---|---|---|---|---|---|---|---|---|---|
| | $\Xi_{\max}$ | $1/2\,\Xi_{\max}$ | $1/4\,\Xi_{\max}$ | $\Xi_{\max}$ | $1/2\,\Xi_{\max}$ | $1/4\,\Xi_{\max}$ | $\Xi_{\max}$ | $1/2\,\Xi_{\max}$ | $1/4\,\Xi_{\max}$ | $\Xi_{\max}$ | $1/2\,\Xi_{\max}$ | $1/4\,\Xi_{\max}$ |
| n=16 | $51.22 \pm 0.08$ | $51.79 \pm 0.10$ | $51.71 \pm 0.03$ | $51.59 \pm 0.02$ | $51.67 \pm 0.01$ | $51.65 \pm 0.13$ | $51.80 \pm 0.10$ | $51.81 \pm 0.13$ | $51.81 \pm 0.04$ | $51.72 \pm 0.02$ | $51.76 \pm 0.01$ | $51.74 \pm 0.06$ |
| n=32 | $50.76 \pm 0.18$ | $51.27 \pm 0.07$ | $51.60 \pm 0.21$ | $51.39 \pm 0.07$ | $51.59 \pm 0.04$ | $51.66 \pm 0.12$ | $51.79 \pm 0.06$ | $51.73 \pm 0.10$ | $51.77 \pm 0.10$ | $51.70 \pm 0.02$ | $51.71 \pm 0.02$ | $51.70 \pm 0.02$ |

Table 4: **Quality propagation across training phases with different consensus distances** on ResNet-20 for CIFAR-10 (Ring with $n=32$). In phase-1 and phase-2, the model parameters reach inexact consensus controlled by different target consensus distance $\Xi$, while phase-3 performs All-Reduce on model parameters.

| phase-2  phase-1 | $\Xi_{\max}$ | $1/2\,\Xi_{\max}$ | $1/4\,\Xi_{\max}$ |
|---|---|---|---|
| $1/2\,\Xi_{\max}$ | $92.48 \pm 0.19$ | $92.46 \pm 0.11$ | $92.31 \pm 0.23$ |
| $1/4\,\Xi_{\max}$ | $92.73 \pm 0.11$ | $92.66 \pm 0.08$ | $92.69 \pm 0.19$ |
| $1/8\,\Xi_{\max}$ | $93.10 \pm 0.22$ | $92.88 \pm 0.15$ | $92.91 \pm 0.06$ |

Table 5: **The impact of different numbers of training epochs (at phase-1)** on generalization, for training ResNet-20 on CIFAR-10 (dec-phase-1 with $n = 32$). The number of epochs at phase-1 is chosen from $\{150, 200, 250\}$, while the rest of the training reuses our default setup.

| target Ξ | training epochs at phase-1 | | |
|---|---|---|---|
| | 150 | 200 | 250 |
| $\Xi_{\max}$ | $91.78 \pm 0.35$ | $91.91 \pm 0.19$ | $92.04 \pm 0.14$ |
| $1/2\,\Xi_{\max}$ | $92.36 \pm 0.21$ | $92.55 \pm 0.07$ | $92.67 \pm 0.13$ |
| $1/4\,\Xi_{\max}$ | $92.74 \pm 0.10$ | $92.91 \pm 0.15$ | $92.84 \pm 0.20$ |

sensible in the initial training phase: *there exists a critical consensus distance for the training, beyond which the impact of decentralization is negligible.*

**A non-negligible consensus distance at middle phases can improve generalization over centralized training.** Surprisingly, it is not always the case that the generalization performance improves with the shrinking consensus distance. We observe that at the phase right after initial training plateaus (e.g. phase-2 for CIFAR-10, phase-3 for Imagenet-32), a non-negligible consensus distance[8] actually boosts the generalization performance over the centralized training which has been deemed optimal. In CIFAR-10 dec-phase-2 experiments (Table 2), the generalization performance increases monotonically with the evaluated consensus distance and is consistently superior to that of the centralized training (e.g. 93.04, 92.99, 92.87 over 92.82 for $n = 32$). Analogous observation can be obtained in Imagenet-32 dec-phase-3 experiments (Table 3).

It coincides with the observations firstly introduced in post-local SGD (Lin et al., 2020b), where for better generalization, consensus distance is created among local machines by less frequent model parameter synchronization (All-Reduce) in late training phases (e.g. phase-2, phase-3 for CIFAR). Thus non-negligible consensus distance at middle phases can be viewed as a means of injecting proper noise as argued in Lin et al. (2020b), which reduces communication cost and in the meanwhile benefits generalization.

**At the last phase of training, the consensus degree marginally impacts the generalization.** Similar to the initial training phase, the final convergence phase seems to favor small consensus distances in CIFAR-10 experiments. However, its impact is less prominent in comparison: for dec-phase-3, performance of a smaller consensus distance ($1/4\,\Xi_{\max}$) is only $0.25\%$ and $0.37\%$ higher than that of $\Xi_{\max}$ for $n = 32, 64$ respectively (Table 2). In Imagenet-32 experiments, dec-phase-3 performance is not even affected by changes in consensus.

**Quality propagation across phases.** Our previous experiments only considered a single phase of decentralized training. We now evaluate the continued impact of consensus across the sequence of multiple phases. In Table 4, we control the consensus distance for both phase-1 and phase-2 when training CIFAR-10. When one phase is held under every specific consensus control, our previous findings for the other phase still hold. For instance, when we apply $1/2\,\Xi_{\max}, 1/4\,\Xi_{\max}$ consensus control to phase-2 (middle column in Table 4), we can still observe that a smaller consensus distance in phase-1 results in a higher performance as in our previous finding. Hence our previous findings are valid in more general cases of decentralized training.

---

[8] In Table 17 and 16 in Appendix C.3.1, we show that there exists optimal consensus distance at middle phases, beyond which the gain in generalization (brought by noise injection) starts to diminish.

Table 6: **The importance of phase-1** for training ResNet-20 on CIFAR-10 ($n = 32$), in terms of **(1) target consensus distance** and **(2) the number of training epochs**. In phase-2 and phase-3, we perform decentralized training (w/o consensus distance control).

| # of epochs | target $\Xi$    $\Xi_{max}$ | $1/2\ \Xi_{max}$ | $1/4\ \Xi_{max}$ | $1/8\ \Xi_{max}$ | $0\ \Xi_{max}$ |
|---|---|---|---|---|---|
| 150 | $91.74 \pm 0.15$ | $92.31 \pm 0.12$ | $92.81 \pm 0.22$ | $92.91 \pm 0.15$ | $92.94 \pm 0.07$ |
| 200 | $91.81 \pm 0.22$ | $92.88 \pm 0.20$ | $93.00 \pm 0.18$ | $93.01 \pm 0.10$ | $92.90 \pm 0.17$ |
| 250 | $92.09 \pm 0.23$ | $92.74 \pm 0.11$ | $93.15 \pm 0.26$ | $92.99 \pm 0.24$ | $93.31 \pm 0.06$ |

**Longer training cannot close the generalization gap caused by large consensus distances in initial training phase.** As discussed above, large consensus distances in the initial phase can result in significant generalization loss due to the optimization difficulty. Table 5 investigates whether a prolonged training on the initial phase can address this difficulty: we prolong the phase-1 for CIFAR-10 with a range of consensus distances and leave the other training phases centralized. We can observe that although longer training is beneficial for each consensus distance, it cannot recover the generalization gap resulting from large consensus distance. For instance, the maximum gain (among all evaluated cases) of increasing the epoch number from 150 to 250 is $0.31\%$ at $1/2\ \Xi_{max}$, which is lower than the average gain (around $0.6\%$) of merely reducing the consensus distance from $\Xi_{max}$ to $1/2\ \Xi_{max}$. Table 13 evaluates cases where dec-phase-2 and dec-phase-3 are prolonged. We find that longer training in these two phases bring about negligible performance gain.

**Practical guidelines: prioritizing the initial training phase.** Apart from effectiveness (generalization/test performance), efficiency (time) stands as the other crucial goal in deep learning, and thus how to allocate communication resource over the training becomes a relevant question.

As indicated by our first empirical finding (and theory in Section 3), initial training phase bears the greatest importance over all other training phases; therefore the communication expenditure should be concentrated on the initial phase to maintain a consensus distance lower than the CCD. A list of communication topologies with superior spectral properties, e.g. exponential graph (Assran et al., 2019) and random matching (Nadiradze et al., 2020), can be utilized[9] to achieve fast convergence in gossip averaging.

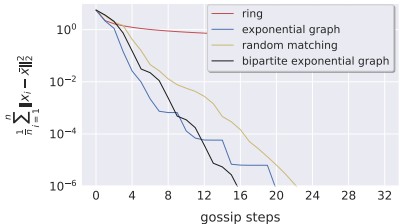

Figure 3: Understanding the consensus averaging problem for different communication topologies ($n = 32$). Results on different communication scales are deferred to Appendix C.1.

The late training phases should be less prioritized for communication resources, due to the generalization benefits from a reasonable consensus distance in the middle phases. Providing a rigorous way to quantify the optimal consensus distance is non-trivial, and we leave it as future work.

In Table 6 we show that the above mentioned guideline is practically feasible: as long as the quality of the initial phase is ensured, we can afford to slacken the consensus control for later phases, in particular the middle phase. For instance, when the number of epochs is 150, a consensus control of $1/4\ \Xi_{max}$ in the initial phase with uncontrolled middle and final phase is adequate to recover the centralized training performance (92.81 v.s. 92.82). Note that here the noise injection from the uncontrolled middle phase also contributes positively to the performance. Table 16 in Appendix C.3.1 additionally justifies the applicability of applying this guideline on exponential graph, and we leave more complicated communication scheme design as future work.

## 4.4 PRELIMINARY STUDY ON TRAINING TRANSFORMER MODELS

**The critical consensus distance also exists in NLP tasks.** Figure 4(c) demonstrates that $1/4\ \Xi_{max}$ target control on a ring is sufficient to recover the centralized training performance. Besides, the target consensus distance in this case can be reached by exponential graph (and thus target test performance, as shown in Figure 4(a) and 4(b)). These justify the importance of designing an efficient communication topology/scheme in practice so as to effectively reach the critical consensus distance.

---

[9] The definition of the communication topology is detailed in Appendix C.1.

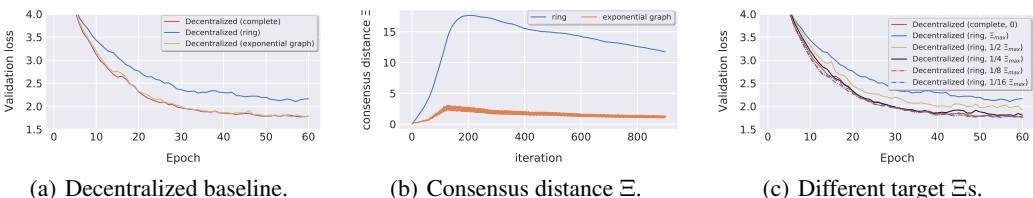

| (a) Decentralized baseline. | (b) Consensus distance $\Xi$. | (c) Different target $\Xi$s. |

Figure 4: Learning curves for training Transformer on Multi30k ($n=32$).

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

# A  THEORY

In this section, we prove the claims from Section 3.

## A.1  PROOF OF REMARK 2, CRITICAL CONSENSUS DISTANCE

The proof of this claim follows by the following Lemma:

**Lemma 5** (Koloskova et al. (2020b), Descent lemma for non-convex case). *Under the given assumptions, and for any stepsize $\gamma < \frac{1}{4L}$, the iterates of D-SGD satisfy*

$$\mathbb{E}_{t+1} \, f(\bar{\mathbf{x}}^{(t+1)}) \leq f(\bar{\mathbf{x}}^{(t)}) - \frac{\eta}{4} \left\| \nabla f(\bar{\mathbf{x}}^{(t)}) \right\|_2^2 + \gamma \Xi_t^2 + \frac{L}{n} \gamma^2 \hat{\sigma}^2.$$

*Proof.* By replacing $\Xi_t$ in the above inequality with (4), we simplify:

$$\mathbb{E}_{t+1} \, f(\bar{\mathbf{x}}^{(t+1)}) \leq f(\bar{\mathbf{x}}^{(t)}) - \frac{\eta}{8} \left\| \nabla f(\bar{\mathbf{x}}^{(t)}) \right\|_2^2 + \frac{2L}{n} \gamma^2 \hat{\sigma}^2.$$

This inequality now matches (up to differences in the constants) the standard recursion that one can derive for C-SGD (Dekel et al., 2012; Bottou et al., 2018; Stich & Karimireddy, 2019). □

## A.2  PROOF OF PROPOSITION 3, TYPICAL CONSENSUS DISTANCE

We need an auxiliary (but standard) lemma, to estimate the change of the consensus distance between iterations.

**Lemma 6** (Consensus distance). *It holds*

$$\Xi_{t+1}^2 \leq (1 - p/2)\Xi_t^2 + \frac{3(1-p)\gamma^2}{p} \left( \phi_t^2 + p\sigma^2 \right).$$

We give the proof of this statement shortly below. First, let us consider how this lemma allows the proof of the claim. For this, we first consider a particular special case, and assume $\phi_t \leq \phi$, for a constant $\phi$. In this case, we can easily verify by unrolling the recursion:

$$\Xi_t^2 \leq \sum_{i=0}^{t-1} (1-p/2)^i \frac{3(1-p)\gamma^2(\phi^2 + p\sigma^2)}{p} \leq 6(1-p)\gamma^2 \left( \frac{\phi^2}{p^2} + \frac{\sigma^2}{p} \right).$$

Now, for the claim in the main text, we use assumption that $\phi_t$ are changing slowly, that is, not decreasing faster than exponentially: $\phi_t^2 \leq (1 + p/4)\phi_{t+1}^2$. With this assumption, and observing $(1 - p/2)^i (1 + p/4)^i \leq (1 - p/4)^i$, we can unroll as before

$$\Xi_t^2 \leq \sum_{i=0}^{t-1} (1-p/2)^i \frac{3(1-p)\gamma^2(\phi_{t-i-1}^2 + p\sigma^2)}{p}$$

$$\leq \sum_{i=0}^{t-1} (1-p/4)^i \frac{3(1-p)\gamma^2(\phi_{t-1}^2 + p\sigma^2)}{p} \leq 12(1-p)\gamma^2 \left( \frac{\phi_{t-1}^2}{p^2} + \frac{\sigma^2}{p} \right).$$

*Proof of Lemma 6.* We use the following matrix notation here

$$\mathbf{X}^{(t)} := \left[ \mathbf{x}_1^{(t)}, \dots, \mathbf{x}_n^{(t)} \right] \in \mathbb{R}^{d \times n},$$

$$\bar{\mathbf{X}}^{(t)} := \left[ \bar{\mathbf{x}}^{(t)}, \dots, \bar{\mathbf{x}}^{(t)} \right] = \mathbf{X}^{(t)} \frac{1}{n} \mathbf{1} \mathbf{1}^\top,$$

$$\nabla F(\mathbf{X}^{(t)}, \xi^{(t)}) := \left[ \nabla F_1(\mathbf{x}_1^{(t)}, \xi_1^{(t)}), \dots, \nabla F_n(\mathbf{x}_n^{(t)}, \xi_n^{(t)}) \right],$$

$$\nabla f(\mathbf{X}^{(t)}) := \left[ \nabla f_1(\mathbf{x}_1^{(t)}), \dots, \nabla f_n(\mathbf{x}_n^{(t)}) \right].$$

As a reminder, $\Xi_t^2 := \frac{1}{n} \sum_{i=1}^n \left\| \bar{\mathbf{x}}^{(t)} - \mathbf{x}_i^{(t)} \right\|^2$, and $\phi_t^2 := \frac{1}{n} \sum_{i=1}^n \left\| \nabla f_i(\mathbf{x}_i^{(t)}) \right\|^2$.

$$
\begin{aligned}
n\Xi_{t+1}^2 &= \left\| \bar{\mathbf{X}}^{(t+1)} - \mathbf{X}^{(t+1)} \right\|_F^2 = \left\| (\mathbf{X}^{(t)} - \gamma \nabla F(\mathbf{X}^{(t)}, \xi^{(t)})) \left( \frac{1}{n} \mathbf{1}\mathbf{1}^\top - \mathbf{W} \right) \right\|_F^2 \\
&= \left\| (\mathbf{X}^{(t)} - \gamma \nabla F(\mathbf{X}^{(t)}, \xi^{(t)})) \left( \frac{1}{n} \mathbf{1}\mathbf{1}^\top - \mathbf{I} \right) \left( \frac{1}{n} \mathbf{1}\mathbf{1}^\top - \mathbf{W} \right) \right\|_F^2 \\
&\leq (1-p) \left\| (\mathbf{X}^{(t)} - \gamma \nabla F(\mathbf{X}^{(t)}, \xi^{(t)})) \left( \frac{1}{n} \mathbf{1}\mathbf{1}^\top - \mathbf{I} \right) \right\|_F^2 \\
&\leq (1-p) \left\| (\mathbf{X}^{(t)} - \gamma \nabla f(\mathbf{X}^{(t)})) \left( \frac{1}{n} \mathbf{1}\mathbf{1}^\top - \mathbf{I} \right) \right\|_F^2 + (1-p)\gamma^2 \left\| \nabla f(\mathbf{X}^{(t)}) - \nabla F(\mathbf{X}^{(t)}, \xi^{(t)}) \right\|_F^2 \\
&\leq (1-p)(1+\alpha) \left\| \mathbf{X}^{(t)} \left( \frac{1}{n} \mathbf{1}\mathbf{1}^\top - \mathbf{I} \right) \right\|_F^2 + (1-p)(1+\alpha^{-1})\gamma^2 \left\| \nabla f(\mathbf{X}^{(t)}) \right\|_F^2 + (1-p)\gamma^2\sigma^2 n \\
&\overset{\alpha = \frac{p}{2}}{\leq} \left( 1 - \frac{p}{2} \right) n\Xi_t^2 + \frac{3(1-p)}{p}\gamma^2 \left\| \nabla f(\mathbf{X}^{(t)}) \right\|_F^2 + (1-p)\gamma^2\sigma^2 n \qquad \square
\end{aligned}
$$

### A.3 SUFFICIENT BOUNDS TO MEET CRITICAL CONSENSUS DISTANCE

In this section, we show that the claimed bounds in Section 3.3 are sufficient conditions to reach the CCD.

According to Remark 3, there exists an absolute constant $C$, (w.l.o.g. $C \geq 2$) such that

$$
\Xi_t^2 \leq C(1-p)\gamma^2 \left( \frac{\phi_t^2}{p^2} + \frac{\sigma^2}{p} \right)
$$

By smoothness,

$$
\begin{aligned}
\phi_t^2 &= \frac{1}{n} \sum_{i=1}^n \left\| \nabla f_i(\mathbf{x}_i^{(t)}) \right\|^2 \\
&\leq \frac{3}{n} \sum_{i=1}^n \left\| \nabla f_i(\mathbf{x}_i^{(t)}) - \nabla f(\mathbf{x}_i^{(t)}) \right\|^2 + \frac{3}{n} \sum_{i=1}^n \left\| \nabla f(\mathbf{x}_i^{(t)}) - \nabla f(\bar{\mathbf{x}}^{(t)}) \right\|^2 + \frac{3}{n} \sum_{i=1}^n \left\| \nabla f(\bar{\mathbf{x}}^{(t)}) \right\|^2 \\
&\leq 3\zeta^2 + 3L^2\Xi_t^2 + 3 \left\| \nabla f(\bar{\mathbf{x}}^{(t)}) \right\|^2 .
\end{aligned}
$$

Supposing $(1-p)\gamma^2 \leq \frac{p^2}{6CL^2}$, we can therefore estimate

$$
\begin{aligned}
\Xi_t^2 &\leq C(1-p)\gamma^2 \left( \frac{3 \left\| \nabla f(\bar{\mathbf{x}}^{(t)}) \right\|^2 + 3L^2\Xi_t^2 + 3\zeta^2}{p^2} + \frac{\sigma^2}{p} \right) \\
&\leq 3C(1-p)\gamma^2 \left( \frac{\left\| \nabla f(\bar{\mathbf{x}}^{(t)}) \right\|^2 + \zeta^2}{p^2} + \frac{\sigma^2}{p} \right) + \frac{1}{2}\Xi_t^2
\end{aligned}
$$

and hence

$$
\Xi_t^2 \leq 6C(1-p)\gamma^2 \left( \frac{\left\| \nabla f(\bar{\mathbf{x}}^{(t)}) \right\|^2}{p^2} + \frac{\zeta^2}{p^2} + \frac{\sigma^2}{p} \right) \tag{6}
$$

The claimed bounds can now easily be verified, by plugging the provided values into (6). For simplicity in the main text we assume that $\zeta = 0$ (we are in the datacenter training scenario).

**Small $\gamma$.** By choosing $\gamma \leq \frac{p}{4nLC}$, we check that our previous constraint $\gamma^2 \overset{C \geq 2}{\leq} \frac{p^2}{6CL^2}$ is satisfied, and

$$
(6) \leq \frac{\left\| \nabla f(\bar{\mathbf{x}}^{(t)}) \right\|^2}{4n^2 CL^2} + \frac{\gamma\sigma^2}{nL} \overset{C \geq 2}{\leq} (4)
$$

**Small $p$.** By choosing $1 - p \leq \frac{1}{5C(1+\gamma Ln)}$, we note that $p \overset{C \geq 2}{\geq} \frac{9}{10}$. Moreover, our previous constraint $(1-p)\gamma^2 \leq \frac{\gamma^2}{5C} \leq \frac{p^2}{6L^2C}$ is satisfied (note that $\gamma \leq \frac{1}{4L}$ throughout). Hence

$$(6) \leq \frac{4\gamma^2}{5(1+\gamma Ln)}\left(\frac{100\left\|\nabla f(\bar{\mathbf{x}}^{(t)})\right\|^2}{81} + \frac{10\sigma^2}{9}\right) \overset{\gamma \leq 1/(4L)}{\leq} (4)$$

In the above calculations we for the simplicity assumed that $\zeta = 0$. For the general non-iid data case when $\zeta > 0$ we can calculate similar bounds on $\gamma, p$. These bounds would have similar dependence on parameters, and would be stricter. Indeed, the typical consensus distance would be also influenced by non-iidness of the data $\zeta$ and it is therefore harder to satisfy the CCD condition.

### A.4 PROOF OF LEMMA 4, REPEATED GOSSIP

By the assumption stated in the lemma, it holds for each component $\mathbf{W}_i$ of the product $\mathbf{W} = \mathbf{W}_k \ldots \mathbf{W}_1$, $i \in [1, k]$ that

$$\mathbb{E}_{\mathbf{W}_i}\left\|\mathbf{X}\mathbf{W}_i - \bar{\mathbf{X}}\right\|_F^2 \leq (1-p)\left\|\mathbf{X} - \bar{\mathbf{X}}\right\|_F^2, \forall \mathbf{X} \in \mathbb{R}^{d \times n}.$$

Now lets estimate the parameter $p_{\mathbf{W}}$. Using that $\mathbf{W}_i$ are independent

$$\mathbb{E}_{\mathbf{W}}\left\|\mathbf{X}\mathbf{W} - \bar{\mathbf{X}}\right\|_F^2 = \mathbb{E}_{\mathbf{W}_1 \ldots \mathbf{W}_k}\left\|\mathbf{X}\mathbf{W}_k \ldots \mathbf{W}_1 - \bar{\mathbf{X}}\right\|_F^2 =$$
$$= \mathbb{E}_{\mathbf{W}_2 \ldots \mathbf{W}_k}\mathbb{E}_{\mathbf{W}_1}\left\|\mathbf{Y}\mathbf{W}_1 - \bar{\mathbf{Y}}\right\|_F^2,$$

where we defined $\mathbf{Y} = \mathbf{X}\mathbf{W}_k \ldots \mathbf{W}_2$ and used that $\mathbf{W}_i \frac{1}{n}\mathbf{1}\mathbf{1}^\top = \frac{1}{n}\mathbf{1}\mathbf{1}^\top$, so

$$\bar{\mathbf{Y}} = \mathbf{X}\mathbf{W}_k \ldots \mathbf{W}_2 \frac{1}{n}\mathbf{1}\mathbf{1}^\top = \mathbf{X}\frac{1}{n}\mathbf{1}\mathbf{1}^\top = \bar{\mathbf{X}}.$$

Therefore,

$$\mathbb{E}_{\mathbf{W}}\left\|\mathbf{X}\mathbf{W} - \bar{\mathbf{X}}\right\|_F^2 \leq (1-p)\mathbb{E}_{\mathbf{W}_2 \ldots \mathbf{W}_k}\left\|\mathbf{X}\mathbf{W}_k \ldots \mathbf{W}_2 - \bar{\mathbf{X}}\right\|_F^2.$$

Applying the same calculations to the rest, we conclude that $1 - p_{\mathbf{W}} = (1-p)^k$.

## B    DETAILED EXPERIMENTAL SETUP

**Comments on large-batch training.**    Coupling the quality loss issue of the decentralized training with the large-batch training difficulty is non-trivial and is out of the scope of this paper. Instead, we use reasonable local mini-batch sizes (together with the number of workers (denoted as $n$)), as well as the well-developed large-batch training techniques (Goyal et al., 2017), to avoid the difficulty of extreme large-batch training.

**Multi-phase experiment justification.**    The averaged local gradient norm $\phi_t$ as well as the $L$-smoothness of ResNet-20 on CIFAR-10 for a ring and a complete graph ($n = 32$) are shown in Figure 5 and Figure 6 respectively.

The estimation procedure is analogous to that in Santurkar et al. (2018); Lin et al. (2020a): we take 8 additional steps long the direction of current update, each with $0.2$ of normal step size. This is calculated at every 8 training steps. The smoothness is evaluated as the maximum value of $L$ satisfying Assumption 2.

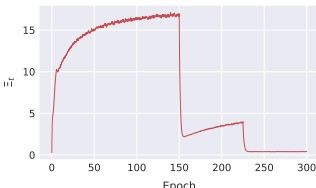 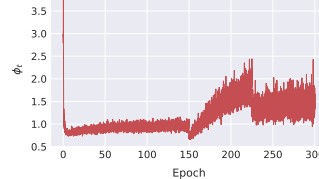 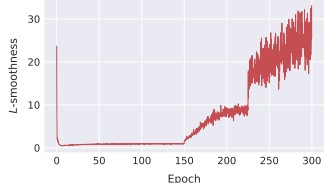

(a) The consensus distance for de-    (b) The averaged norm of the local    (c) The gradient Lipschitz curve for
centralized training.                  gradients for decentralized training.  both decentralized training.

Figure 5: **Justification for our multiple-phase experimental design choice** (on ring graph). We run ResNet-20 on CIFAR-10 ($n\!=\!32$) with the ring topology. We can observe the three quantities most relevant to optimization all naturally form three phases, dictated by the learning rate schedule.

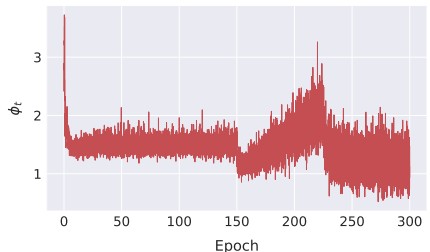 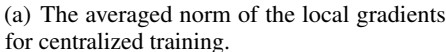 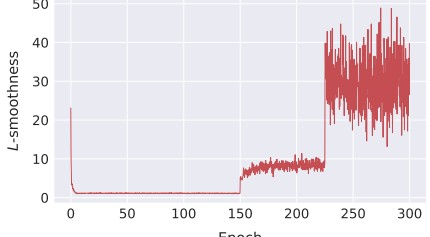

(a) The averaged norm of the local gradients    (b) The gradient Lipschitz curve for both cen-
for centralized training.                        tralized training.

Figure 6: **Justification for our multiple-phase experimental design choice** (on complete graph). We run ResNet-20 on CIFAR-10 ($n\!=\!32$) with the complete topology. We can again observe the three quantities most relevant to optimization all naturally form three phases, dictated by the learning rate schedule.

## C    ADDITIONAL RESULTS

### C.1    UNDERSTANDING ON CONSENSUS AVERAGING PROBLEM

We study a host of communication topologies: (1) deterministic topologies (ring, and complete graph) and (2) undirected time-varying topologies (illustrated below).

- **Random matching** (Boyd et al., 2006). At each communication step, all nodes are divided into non-overlapping pairs randomly. Each node connects all other nodes with equal probability.
- **Exponential graph** (Assran et al., 2019). Each is assigned a rank from 0 to $n - 1$. Each node $i$ periodically communicates with a list nodes with rank $i + 2^0, i + 2^1, \ldots, i + 2^{\lfloor \log_2(n-1) \rfloor}$. In the one-peer-per-node experiments, each node only communicates to one node by cycling through its list. The formed graph is undirected, i.e., both transmission and reception take place in each communication.

- **Bipartite exponential graph** (Lian et al., 2018; Assran et al., 2019). In order to avert dead-locks (Lian et al., 2018), the node with an odd rank $i$ cycles through nodes with even ranks $i + 2^0 - 1, i + 2^1 - 1, \ldots, i + 2^{\lfloor \log_2(n-1) \rfloor} - 1$ by transmitting a message and waiting for a response. while the nodes with even ranks only await messages and reply upon reception.

Table 7 displays the spectral gap and node degree of studied topologies, and Figure 7 provides the convergence curves for different communication topologies on graph scales. Figure 8 in addition visualizes the spectral gap (in expectation) for different communication topologies.

Table 7: Spectral gap and node degree of studied topologies.

| Topologies | Spectral Gaps (in expectation) | Node degrees ($n$ nodes) |
|---|---|---|
| Complete | 1 | $n$ |
| Fixed ring | $\mathcal{O}(\frac{1}{n^2})$ | 2 |
| Exponential graph | $\mathcal{O}(1)$ | 2 |
| Bipartite exponential graph | $\mathcal{O}(1)$ | 1 |
| Random matching | $\mathcal{O}(1)$ | 1 |

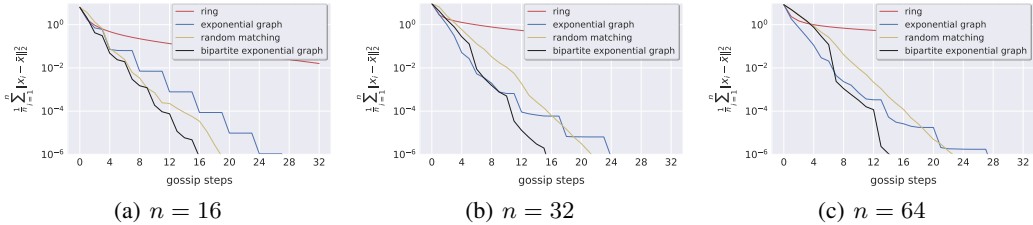

(a) $n = 16$     (b) $n = 32$     (c) $n = 64$

Figure 7: The convergence curves for the consensus averaging problem on different communication topologies and different scales (i.e., $n = 16$, $n = 64$ and $n = 128$). This figure complements the Figure 3 in the main text.

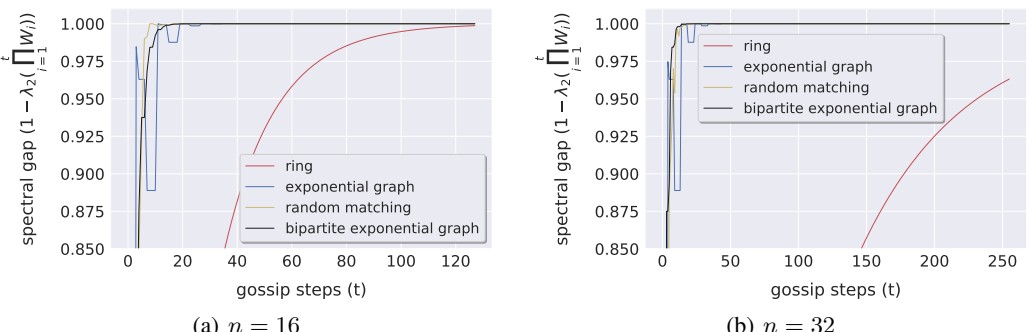

(a) $n = 16$     (b) $n = 32$

Figure 8: The spectral gap (in expectation) of different communication topologies on different graph scales.

Table 8 examines these topologies on a standard deep learning benchmark with different graph scales, while Figure 9 visualizes the required communication rounds (per gradient update step) for a range of consensus distance targets.

| | Complete | Fixed ring | Exponential graph | Bipartite exponential graph | Random matching |
|---|---|---|---|---|---|
| n=16 | $92.91 \pm 0.12$ | $92.51 \pm 0.19$ | $92.63 \pm 0.30$ | $92.76 \pm 0.04$ | $92.65 \pm 0.15$ |
| n=32 | $92.82 \pm 0.27$ | $91.93 \pm 0.05$ | $92.64 \pm 0.04$ | $92.29 \pm 0.15$ | $92.27 \pm 0.17$ |

Table 8: **The effect of communication topologies and scales** (ResNet-20 on CIFAR-10 with $n = 32$). The test top-1 accuracies are over three seeds with fine-tuned learning rates.

## C.2   UNDERSTANDING THE DECENTRALIZED DEEP LEARNING TRAINING FOR CV TASKS

We use ring as our underlying decentralized communication topology in this subsection.

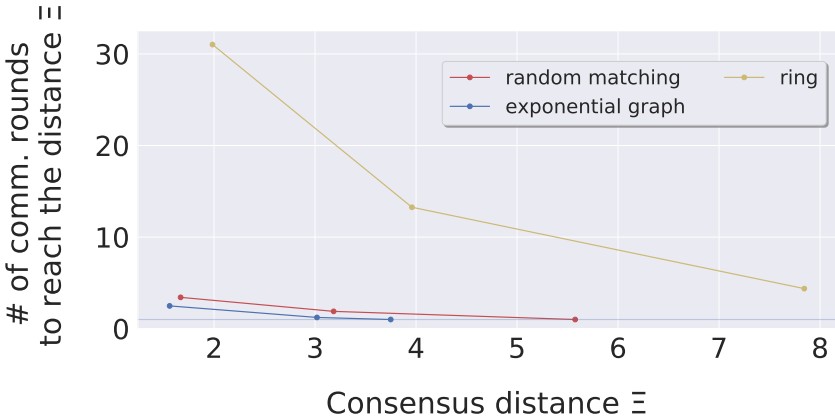

Figure 9: **Target consensus distance v.s. the required communication rounds** (per gradient update step), for training ResNet-20 on CIFAR-10 with different communication topologies. We focus on the setup of dec-phase-1 and vary the target consensus distance for different communication topologies. Due to the changing consensus distance over the training (of the interested phase-1), we consider the averaged consensus distance. The topologies of exponential graph and random matching, empower the capability of fast convergence in gossip averaging and thus only a few steps are required to reach the target consensus distance.

**Elaborated results on consensus distance control.** Table 9 is the elaborated version of Table 2 with more evaluated consensus distances.

Table 9: **The impact of consensus distance of different phases on generalization performance** (test top-1 accuracy) of training ResNet-20 on CIFAR-10. The centralized baseline performance for $n=32$ and $n=64$ are $92.82 \pm 0.27$ and $92.71 \pm 0.11$ respectively. The performance of decentralized training (all phases on a fixed ring and w/o consensus distance control) for $n=32$ and $n=64$ are $91.74 \pm 0.15$ and $89.87 \pm 0.12$ respectively.

| | dec-phase-1 | | | | | dec-phase-2 | | | | dec-phase-3 | | | dec-phase-2 + dec-phase-3 | | |
|---|---|---|---|---|---|---|---|---|---|---|---|---|---|---|---|
| | $\Xi_{max}$ | $1/2\ \Xi_{max}$ | $1/4\ \Xi_{max}$ | $1/8\ \Xi_{max}$ | $1/16\ \Xi_{max}$ | $\Xi_{max}$ | $1/2\ \Xi_{max}$ | $1/4\ \Xi_{max}$ | $1/40\ \Xi_{max}$ | $\Xi_{max}$ | $1/2\ \Xi_{max}$ | $1/4\ \Xi_{max}$ | $\Xi_{max}$ | $1/2\ \Xi_{max}$ | $1/4\ \Xi_{max}$ |
| n=32 | $91.78 \pm 0.35$ | $92.36 \pm 0.21$ | $92.74 \pm 0.10$ | $92.77 \pm 0.25$ | $92.72 \pm 0.05$ | $93.04 \pm 0.01$ | $92.99 \pm 0.30$ | $92.87 \pm 0.11$ | $92.84 \pm 0.27$ | $92.60 \pm 0.00$ | $92.82 \pm 0.21$ | $92.85 \pm 0.24$ | $92.94 \pm 0.07$ | $93.03 \pm 0.24$ | $92.93 \pm 0.15$ |
| n=64 | $90.31 \pm 0.12$ | $92.18 \pm 0.07$ | $92.45 \pm 0.17$ | - | - | $93.14 \pm 0.04$ | $92.94 \pm 0.10$ | $92.79 \pm 0.07$ | - | $92.23 \pm 0.12$ | $92.50 \pm 0.09$ | $92.60 \pm 0.10$ | $92.95 \pm 0.07$ | $92.83 \pm 0.12$ | $92.66 \pm 0.07$ |

**SlowMo cannot fully address the decentralized optimization/generalization difficulty.** Table 10 studies the effectiveness of using SlowMo for better decentralized training. We can witness that even though the performance of decentralized training can be boosted to some extent, it cannot fully address the quality loss issue brought by decentralized training.

Table 10: **The effect of SlowMo for decentralized learning**, for training ResNet20 on CIFAR-10 ($n = 32$). The results (over three random seeds) use the tuned hyper-parameter of SlowMo mentioned in the original paper (Wang et al., 2020). The centralized baseline performance is $92.82 \pm 0.27$.

| topology | w/o SlowMo | w/ SlowMo |
|---|---|---|
| exponential graph | $92.63 \pm 0.22$ | $92.42 \pm 0.36$ |
| ring | $91.74 \pm 0.15$ | $92.53 \pm 0.10$ |

**On the ineffectiveness of tuning learning rate.** Table 11 displays the results of training ResNet-20 on CIFAR-10 (32 nodes), with fine-tuned learning rate on phase-1; learning rate tuning cannot address the test quality loss issue caused by the large consensus distance (i.e. over the CCD).

Table 11: **Phase-1 consensus distance control performance with fine-tuned learning rates** of training ResNet-20 on CIFAR-10 ($n = 32$). Setup in this table is identical to that of Table 2, except that we fine-tune the learning rate for each case from a grid of linear scaling-up factors $\{30, 28, 26, 24, 22\}$. The results are over three seeds.

| | $\Xi_{max}$ | $1/2\ \Xi_{max}$ | $1/4\ \Xi_{max}$ |
|---|---|---|---|
| w/ tuned lr from the search grid | $91.95 \pm 0.26$ | $92.35 \pm 0.24$ | $92.54 \pm 0.08$ |
| w/ default lr | $91.78 \pm 0.35$ | $92.36 \pm 0.21$ | $92.74 \pm 0.10$ |

**Distance control on SGD without momentum.** Table 12 contains the consensus distance control results for SGD without momentum. We can observe a consistent pattern as in Table 2.

Table 12: **The impact of consensus distance on generalization performance with vanilla SGD (without momentum)** (test top-1 accuracy) of training ResNet-20 on CIFAR-10. The All-Reduce performance for $n = 32$ and $n = 64$ are $90.64 \pm 0.19$ and $90.58 \pm 0.26$ respectively. The tuned decentralized performance (all phases on a fixed ring and w/o consensus distance control) for $n = 32$ and $n = 64$ are $90.30 \pm 0.14$ and $88.92 \pm 0.23$ respectively. We repeat experiments for $n = 32$ for 3 seeds and $n = 64$ for 2 seeds.

| target $\Xi$ 
 # nodes | dec-phase-1 | | | dec-phase-2 | | |
|---|---|---|---|---|---|---|
| | $\Xi_{\text{max}}$ | $1/2\Xi_{\text{max}}$ | $1/4\Xi_{\text{max}}$ | $\Xi_{\text{max}}$ | $1/2\Xi_{\text{max}}$ | $1/4\Xi_{\text{max}}$ |
| $n = 32$ | $90.51 \pm 0.05$ | $90.74 \pm 0.14$ | $90.88 \pm 0.37$ | $90.64 \pm 0.18$ | $90.55 \pm 0.19$ | $90.57 \pm 0.17$ |
| $n = 64$ | $88.8 \pm 0.03$ | $89.89 \pm 0.03$ | $90.43 \pm 0.05$ | $90.63 \pm 0.37$ | $90.46 \pm 0.15$ | $90.63 \pm 0.25$ |

**Prolonged training for dec-phase-2 and dec-phase-3.** Table 13 shows the results for prolonged dec-phase-2 and dec-phase-3 on CIFAR-10 with ResNet20. We can observe although longer training duration increases the performance, the improvement is rather small.

Table 13: **The impact of different numbers of training epochs (at phase-2 and phase-3)** on generalization, for training ResNet-20 on CIFAR-10 (ring topology with $n = 32$). The number of epochs at phase-1 is chosen from $\{75, 100, 125\}$, while the rest of the training reuses our default setup. Experiments are run over 2 seeds.

| target $\Xi$ 
 # nodes | dec-phase-2 | | | dec-phase-3 | | |
|---|---|---|---|---|---|---|
| | $\Xi_{\text{max}}$ | $1/2\ \Xi_{\text{max}}$ | $1/4\ \Xi_{\text{max}}$ | $\Xi_{\text{max}}$ | $1/2\ \Xi_{\text{max}}$ | $1/4\ \Xi_{\text{max}}$ |
| 75 epochs | $93.04 \pm 0.01$ | $92.99 \pm 0.30$ | $92.87 \pm 0.11$ | $92.60 \pm 0.00$ | $92.82 \pm 0.21$ | $92.85 \pm 0.24$ |
| 100 epochs | $93.08 \pm 0.08$ | $93.05 \pm 0.16$ | $92.94 \pm 0.03$ | $92.86 \pm 0.16$ | $92.90 \pm 0.18$ | $92.93 \pm 0.19$ |
| 125 epochs | $93.19 \pm 0.16$ | $93.11 \pm 0.17$ | $93.06 \pm 0.07$ | $92.87 \pm 0.23$ | $92.99 \pm 0.25$ | $92.97 \pm 0.20$ |

**The impact of half cosine learning rate schedule.** Table 14 examines the existence of the critical consensus distance with half cosine learning schedule (this scheme is visited in He et al. (2019) as a new paradigm for CNN training). We can witness from Table 14 that the effect of critical consensus distance can be generalized to this learning rate schedule: there exists a critical consensus distance in the initial training phase (as revealed in the inline Figure of Table 14) and ensures good optimization and generalization.

Table 14: **The impact of half cosine learning rate schedule** on generalization, for training ResNet20 on CIFAR-10 (ring topology with $n = 32$). The inline figure depicts the uncontrolled consensus distance over the whole training procedure through the half-cosine learning rate schedule. Only one training phase is considered for the consensus distance control and the numerical results in the table are averaged over 3 seeds.

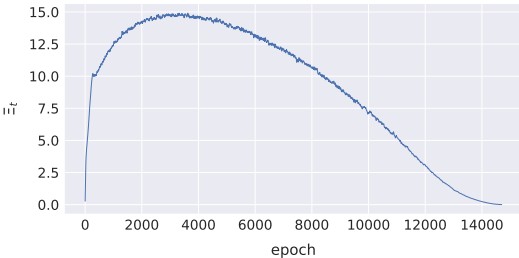

| Ring ($\Xi_{\text{max}}$) | Ring ($1/2\Xi_{\text{max}}$) | Ring ($1/4\Xi_{\text{max}}$) | Ring ($1/8\Xi_{\text{max}}$) | Complete |
|---|---|---|---|---|
| $92.10 \pm 0.06$ | $92.40 \pm 0.10$ | $92.83 \pm 0.11$ | $92.78 \pm 0.05$ | $92.84 \pm 0.22$ |

### C.2.1 ADAPTIVE CONSENSUS DISTANCE CONTROL

In Table 15, we apply the adaptive consensus distance control in the experiments. The observations are consistent with those in constant consensus distance control experiments.

Table 15: **The impact of different consensus distances on optimization and/or generalization, for different phases** of training ResNet-20 on CIFAR-10 ($n = 32$). The table is almost identical to Table 2, except the consensus distance is controlled by the (runtime) averaged norm of the local gradients (i.e. adaptive consensus distance).

|  | $\Xi_{\max}$ | $4\phi_t^{\text{ema}}$ | $2\phi_t^{\text{ema}}$ | $\phi_t^{\text{ema}}$ | $0.5\phi_t^{\text{ema}}$ |
|---|---|---|---|---|---|
| Phase 1 | $91.78 \pm 0.35$ | $91.65 \pm 0.31$ | $92.47 \pm 0.18$ | $92.63 \pm 0.04$ | $92.80 \pm 0.16$ |
| Phase 2 | $93.04 \pm 0.01$ | $93.05 \pm 0.18$ | $93.01 \pm 0.03$ | $93.03 \pm 0.08$ | $92.95 \pm 0.10$ |
| Phase 3 | $92.94 \pm 0.07$ | $92.87 \pm 0.18$ | $92.83 \pm 0.20$ | - | - |

### C.3 CONSENSUS CONTROL WITH OTHER TOPOLOGIES

In Table 16, we exert consensus control with an exponential graph as the base communication topology. We can observe that our findings from main experiments with a ring base topology are valid.

Table 16: The **impact of quality propagation across phases** (in both phase 1 and phase 2) on an **undirected time-varying exponential graph** ($n = 32$), similar to Table 4.

| phase 1 \ phase 2 | local update step = 1 | | | | local update step = 2 | | | | local update step = 4 | | | |
|---|---|---|---|---|---|---|---|---|---|---|---|---|
|  | $\Xi_{\max}$ | $2\phi_t^{\text{ema}}$ | $\phi_t^{\text{ema}}$ | $0.5\phi_t^{\text{ema}}$ | $\Xi_{\max}$ | $2\phi_t^{\text{ema}}$ | $\phi_t^{\text{ema}}$ | $0.5\phi_t^{\text{ema}}$ | $\Xi_{\max}$ | $2\phi_t^{\text{ema}}$ | $\phi_t^{\text{ema}}$ | $0.5\phi_t^{\text{ema}}$ |
| $2\phi_t^{\text{ema}}$ | $92.43 \pm 0.16$ | $92.44 \pm 0.24$ | $92.36 \pm 0.06$ | $92.45 \pm 0.01$ | - | - | - | - | - | - | - | - |
| $1\phi_t^{\text{ema}}$ | $92.58 \pm 0.09$ | $92.37 \pm 0.14$ | $92.63 \pm 0.09$ | $92.51 \pm 0.16$ | - | - | - | - | - | - | - | - |
| $0.5\phi_t^{\text{ema}}$ | $92.74 \pm 0.17$ | $92.56 \pm 0.19$ | $92.56 \pm 0.21$ | $92.75 \pm 0.24$ | $92.79 \pm 0.13$ | $92.68 \pm 0.21$ | $92.65 \pm 0.07$ | $92.68 \pm 0.22$ | $92.85 \pm 0.09$ | $92.76 \pm 0.09$ | $92.72 \pm 0.21$ | $92.75 \pm 0.09$ |
| $0.25\phi_t^{\text{ema}}$ | $92.71 \pm 0.13$ | $92.72 \pm 0.08$ | $92.81 \pm 0.20$ | $92.76 \pm 0.24$ | $92.83 \pm 0.21$ | $92.86 \pm 0.16$ | $92.86 \pm 0.13$ | $92.81 \pm 0.26$ | $93.13 \pm 0.09$ | $92.88 \pm 0.16$ | $92.85 \pm 0.26$ | $92.77 \pm 0.23$ |

### C.3.1 THE EXISTENCE OF THE OPTIMAL CONSENSUS DISTANCE FOR NOISE INJECTION.

Table 17 uses a different communication topology (i.e. time-varying exponential graph) for decentralized optimization. Here exponential graph with large spectral gap is applied to CIFAR-10 dec-phase-2 training. We apply the adaptive consensus distance control in this set of experiments. We can observe that increasing consensus distance further by taking local steps improves generalization, however, too many local steps diminish the performance. For instance, for ratio=2, the performance peaks at local update steps 2 and drops at local update 4. It points out that an optimal consensus distance is required to inject proper stochastic noise for better generalization.

Table 17: **The impact of different consensus distances at phase 2**, for training ResNet-20 on CIFAR-10 with time-varying exponential graph ($n = 32$). The baseline performance of using exponential graph for the entire decentralized training is $92.64 \pm 0.04$. The reported test top-1 accuracies are averaged over three seeds.

| local update step = 1 | | | | local update step = 2 | | | local update step = 4 | | |
|---|---|---|---|---|---|---|---|---|---|
| $\Xi_{\max}$ | $2\phi_t^{\text{ema}}$ | $\phi_t^{\text{ema}}$ | $0.5\phi_t^{\text{ema}}$ | $2\phi_t^{\text{ema}}$ | $\phi_t^{\text{ema}}$ | $0.5\phi_t^{\text{ema}}$ | $2\phi_t^{\text{ema}}$ | $\phi_t^{\text{ema}}$ | $0.5\phi_t^{\text{ema}}$ |
| $92.83 \pm 0.12$ | $92.80 \pm 0.09$ | $92.74 \pm 0.27$ | $92.77 \pm 0.19$ | $93.04 \pm 0.08$ | $92.85 \pm 0.17$ | $92.80 \pm 0.02$ | $92.87 \pm 0.10$ | $92.90 \pm 0.12$ | $92.88 \pm 0.19$ |

### C.4 RESULTS FOR TRAINING TRANSFORMER ON MULTI30K

We additionally report the decentralized training results, for a downsampled transformer models (by the factor of 2 w.r.t. the base model in Vaswani et al. (2017)) on Multi30k (Elliott et al., 2016). Figure 10 shows that the straightforward application of Adam in the decentralized manner does encounter generalization problems, which are attributed to the fact that the different local moment buffers (in addition to the weights) become too diverse. Tuning the learning rate schedule cannot address the issue of decentralized Adam, as shown in the Figure 10(b).

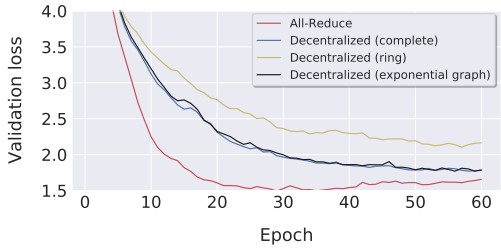 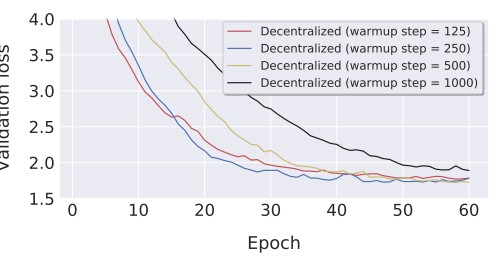

(a) The limitation of decentralized learning with Adam, caused by the different local moment buffers.

(b) Tuning the learning rate cannot alleviate the issue of decentralized Adam.

Figure 10: **Learning curves for training the transformer model on the Multi30k dataset** ($n = 32$). In Figure 10(b), we tune the the number of warmup steps as as way of tuning the learning rate, as the learning rate used in transformer training (Vaswani et al., 2017) is deterministically controlled by the model's dimensionality, the current step index, and the number of warmup steps.

