# OpenReview forum: "On the Effect of Consensus in Decentralized Deep Learning"
_ICLR.cc/2021/Conference — Reject_

### Official Review · AnonReviewer2 · 2020-10-24
**Useful contributions on decentralized methods for training deep network, if somewhat incremental**

**Rating:** 7
**Confidence:** 5

**Review:**

This paper studies decentralized gradient methods for training deep networks. It focuses on the so-called "critical consensus distance" and how disagreement during different stages of training ultimately effects optimization (training loss) and learning (generalization error). Theory is provided for the case of synchronous symmetric averaging methods, and the paper is complemented with detailed experiments on CIFAR and tiny-ImageNet.

This is a nice contribution to the growing literature on decentralized training for deep neural networks. The connection between consensus distance and performance has previously been studied to a limited extent in various settings, so the contribution of this work is somewhat incremental. However, this paper makes the connection somewhat more rigorous through the theoretical developments in Section 3, and it provides a more detailed empirical investigation than previous work. I expect the results to be useful to those working on decentralized training and am supportive of accepting it.

I have a few suggestions and comments, about which I look forward to hearing from the authors.
1. You mention that consensus distance has previously been investigated to some extent (e.g., Fig 2 in Assran et al. 2019). Are there connections between consensus distance and other quantities that have been considered in the literature to relate training to performance (e.g., gradient diversity as in Yin et al. 2017, or the closely related gain ratio in Johnson et al., 2020). Similarly, is there a connection to stochastic weight averaging (Izmailov et al. 2018) and it's parallel version (Gupta et al., 2020)?
2. It is not clear if there are specific aspects of the tasks considered that are important for the findings to hold. CIFAR-10 and ImageNet-32 are both relatively small datasets. Is it possible that in the centralized setting, ResNet-20 is overfitting, and the error from decentralized SGD has a regularizing effect, leading to better generalization? It would be interesting to perform further experiments to explore if this is the case.
3. It would also be interesting to know if the results similarly carry over to the standard (higher-resolution) ImageNet training and models (e.g., ResNet-50), to know if the phenomena observed are relevant to large-scale training. While I appreciate that CIFAR and ImageNet-32 experiments are useful for quick experimentation, and running experiments on the standard ImageNet task are much more computationally expensive, CIFAR and ImageNet-32 are not very reflective of tasks where one would normally use distributed or decentralized training, since one can easily train a model on them using a single GPU in a reasonable amount of time (~1 hour).
4. Regarding the experiments in Table 5 (longer training), why focus on prolonging training in phase 1? I would expect that extending later phases would potentially allow to overcome issues due to large consensus distances in phase 1. Did you explore this?
5. The analysis focuses on symmetric (push-pull) mixing. Do you expect the same trends to carry over to push-only methods such as those considered in Assran et al., 2019?
6. Nowadays, the half-cosine learning rate schedule is also commonly used for CV tasks (He et al. 2018). How do you expect this to affect CCD and the analysis leading to Remark 2?
7. How does using more gossip iterations impact the practical utility of these methods? In particular, standard implementations of all_reduce only require that each node communicate 2 copies of the parameters per iteration. Now that we need to potentially perform multiple rounds of gossip between each optimizer update, are decentralized methods still attractive for reducing overall training time? On a related note, Tsianos and Rabbat (2014) also proposed to use multiple rounds of gossip to essentially reduce the CCD for convex problems, and show that it can lead to overall less communication overhead to reach a desired level of accuracy. Is it possible to show something similar in this setting?

Additional references mentioned:
* Gupta, Serrano, DeCoste, "Stochastic weight averaging in parallel: Large-batch training that generalizes well," ICLR 2020 and arxiv:2001.02312
* He, Zhang, Zhang, Zhang, Xie, and Li, "Bag of tricks for image classification with convolutional neural networks," CVPR 2019 and arxiv:1812.01187
* Izmailov, Podoprikhin, Garipov, Vetrov, and Wilson, "Averaging weights leads to wider optima and better generalization," arxiv:1803.05407
* Johnson, Agrawal, Gu, and Guestrin, "AdaScale SGD: A user-friendly algorithm for distributed training" ICML 2020 and arxiv:2007.05105
* Tsianos and Rabbat, "Efficient distributed online prediction and stochastic optimization with approximate distributed averaging" IEEE Trans Signal and Information Procesing over Networks 2016 and arxiv:1403.0603
* Yin, Pananjady, Lam, Papailiopoulos, Ramchandran, and Bartlett, "Gradient diversity: A key ingredient for scalable distributed learning," AISTATS 2018 and arxiv: 1706.05699

---

> ### Author Response · Authors · 2020-11-19
> **Response to R2, part 1/2**
>
> We thank the reviewer for the time and valuable feedback. We will add corrections/clarifications as suggested. Please find answers to specific comments below:
>
> ### Connection with gradient diversity (Question 1)
> The connections between the consensus distance and gradient diversity measure are not obvious and is an interesting direction for future works. On the one hand, decentralized methods could suffer from similar problems as centralized ones if gradients are not diverse enough. On the other hand, it is harder to achieve consensus (some constant accuracy epsilon) on the diverse vectors (gradients) rather than on similar ones.
>
> ### Connection with other methods like SWA/SWAP (Question 1)
> Our empirical results share a similar insight as in SWA, SWAP, and Post-local SGD, but none of them consider decentralized learning.
>
> SWA is a method where models are sampled from the later stages of an SGD training run; when the weights of these models are averaged, they result in a model with much better generalization properties.
> SWAP extends SWA in a parallel fashion: it uses a large batch size to train the model close to convergence and then switches to several individual runs with a small mini-batch size. These individual runs serve as a way of sampling from a posterior distribution and can be averaged for better generalization performance (i.e. the idea of SWA).
>
> Post-local SGD, SWA, SWAP, as well as the empirical insights presented in our paper, are closely related: we first need sufficient small consensus distance to guarantee the optimization quality (in post-local SGD, SWA, and SWAP, the consensus distance equals 0) and thus different model averaging choices can be utilized in the later training phase for better generalization. Considering the later training phase, our empirical observations in decentralized learning suggest that we can improve the generalization through the simultaneous SGD with gossip averaging. This is analogous but different from SWA and SWAP that sample model independently (i.e., perform SGD) from the well-trained model and average over sampled models; and similar to Post-local SGD which performs simultaneous SGD steps with infrequent averaging.
>
> ### Experiments on standard ImageNet (Question 2 & 3)
> We conducted experiments on downsampled ImageNet (image resolution 32) with ResNet-20-3 (width factor 3) in Table 3; it has already reached the limit of our computational resources.
>
> Our insights can be generalized to standard ImageNet training or other challenge tasks, as supported by the existing papers. For example, post-local SGD paper shows on standard ImageNet of the effectiveness of performing local SGD on later training phase (our takeaway message 2: a non-negligible consensus distance at middle phase can improve generalization), and Assran et al. 2019 preliminarily present in their Table 3 the result for standard ImageNet, which is also consistent with our insights (our takeaway message 1: critical consensus distance exists in the initial training phase ensures good optimization and generalization).
>
> ### Insights for prolonged training on other phases (Question 4)
> We prolong the training for dec-phase-2 and dec-phase-3 for node n =32. All experiments are performed over two seeds. We can observe although longer training duration increases the performance, the improvement is rather small.
>
> |             	|                 	| $75$ epochs       	| $100$ epochs     	| $125$ epochs     	|
> |-------------	|-----------------	|-------------------	|------------------	|------------------	|
> | dec-phase-2 	| $\Xi_{max}$     	| $93.04 \pm 0.01$  	| $93.08 \pm 0.08$ 	| $93.19 \pm 0.16$ 	|
> |             	| $1/2 \Xi_{max}$ 	| $92.99 \pm 0.30$  	| $93.05 \pm 0.16$ 	| $93.11 \pm 0.17$ 	|
> |             	| $1/4 \Xi_{max}$ 	| $92.87 \pm 0.11$  	| $92.94 \pm 0.03$ 	| $93.06 \pm 0.07$ 	|
> | dec-phase-3 	| $\Xi_{max}$     	| $92.60 \pm 0.00 $ 	| $92.86 \pm 0.16$ 	| $92.87 \pm 0.23$ 	|
> |             	| $1/2 \Xi_{max}$ 	| $92.82 \pm 0.21$  	| $92.90 \pm 0.18$ 	| $92.99 \pm 0.25$ 	|
> |             	| $1/4 \Xi_{max}$ 	| $92.85 \pm 0.24$  	| $92.94 \pm 0.19$ 	| $92.97 \pm 0.20$ 	|

---

> ### Author Response · Authors · 2020-11-19
> **Response to R2, part 2/2**
>
> ### Observations on cosine learning rate schedule (Question 6)
> We include results of consensus distance control with half cosine learning schedule on ring topology (this scheme is visited in [3] as a new paradigm for CNN training).
> We can observe that the effect of critical consensus distance can be generalized to this learning rate schedule:
> there exists a critical consensus distance in the initial training phase (please refer to the inline Figure of Table 14 in the revised paper) ensures good optimization and generalization.
>
> | $\Xi_{max}$      	| $1/2 \Xi_{max}$  	| $1/4 \Xi_{max}$  	| $1/8 \Xi_{max}$  	| Complete         	|
> |------------------	|------------------	|------------------	|------------------	|------------------	|
> | $92.10 \pm 0.06\quad$ 	| $92.40 \pm 0.10\quad$ 	| $92.83 \pm 0.11\quad$ 	| $92.78 \pm 0.05\quad$ 	| $92.84 \pm 0.22$ 	|
>
> [Comments on developing practical algorithms (Question 7)]
> Our work aims to better understand the importance of consensus distance on different phases of deep learning training: performing multiple gossip steps serves as a way of controlling consensus distance for the understanding purpose, rather than as an efficient solution in practice.
>
> We believe our insights can be utilized for efficient and effective algorithm design in decentralized deep learning, e.g. the communication topology design balancing the trade-off between communication efficiency and the spectral gap. For example, most prior works in communication-efficient topology design generally focus on improving the spectral gap of the topology (e.g. random matching idea in [1, 2]), as motivated by standard convergence analysis. Our work identifies the existence of the critical consensus distance and thus relaxes the requirement on the spectral gap: this insight provides more flexibility to design efficient and effective decentralized deep learning algorithms.
>
> We thank the reviewer for pointing out Tsianos and Rabbat (2014), the paper uses multiple gossip steps on dual averaging methods (convex problems). We will include a discussion in the next revision.
>
> ### Reference
> 1. MATCHA: Speeding up Decentralized SGD via Matching Decomposition Sampling.
> 2. SwarmSGD: Scalable Decentralized SGD with Local Updates
> 3. Bag of Tricks for Image Classification with Convolutional Neural Networks.

---

> > ### Comment · AnonReviewer2 · 2020-11-21
> > **Acknowledging responses**
> >
> > Thanks for your responses (specific to my questions and comments, as well as the general response). I will take this all into account. In my view the main limitation is the restriction to smaller tasks (CIFAR10, low-res ImageNet) with small networks. In my own experience, conclusions drawn on these smaller tasks often fail to hold or generalize to larger-scale tasks. However, the paper contains enough other innovations and insights, and the experimental methodology is otherwise rigorous. For now I am not inclined to change my decision (up or down). The paper is worthy of acceptance at ICLR.

---

### Official Review · AnonReviewer4 · 2020-10-27
**Great topic but need more thoughtful discussions**

**Rating:** 6
**Confidence:** 5

**Review:**

The authors consider the decentralized optimization problem and explain the generalization gap using the consensus distance. They show that when the consensus distance does not grow too large, the performance of centralized training can be reached and sometimes surpassed. The conducted experiments are extensive and the delivered message is pretty clear -- Critical consensus distance exists in the initial training phase and ensures good optimization and generalization, while a non-negligible consensus distance at middle phases can improve generalization over centralized training.

On the theory side, the main contribution is Remark 2 and proposition 3, but why remark 2 relates to generalization are unclear (it only shows the convergence rate), neither does proposition 3 (it only shows the consensus distance). How do we relate the convergence rate differences with the generalization capability is unclear. So I would say the abstract is a bit overclaiming, the authors better tune down their claims to practical only, without any theoretical guarantees -- "We identify the changing consensus distance between devices as a key parameter to explain the gap between centralized and decentralized training. We show that when the consensus distance does not grow too large, the performance of centralized training can be reached and sometimes surpassed."

On the literature side, besides the gossip-based decentralized methods, there are also many primal-dual based decentralized optimization methods [1,2]. In those methods, there will be no mixing matrix and hard to run multiple mixing steps, the authors better also comment on those and discuss how the proposed findings can help these works.

Overall speaking, I feel the motivation and message delivering is clear, though I am afraid that the main contribution falls into the practical findings (they are also important though) instead of the theoretical guarantees -- there is a mismatch between theory and implementations.

[1] Mingyi Hong, Davood Hajinezhad, and Ming-Min Zhao. "Prox-PDA: The proximal primal-dual algorithm for fast distributed nonconvex optimization and learning over networks." International Conference on Machine Learning. 2017.
[2] Haoran Sun and Mingyi Hong. "Distributed non-convex first-order optimization and information processing: Lower complexity bounds and rate optimal algorithms." IEEE Transactions on Signal Processing 67.22 (2019): 5912-5928.

------
update after rebuttal

After reading the author's response, the authors stated that they indeed identify the optimization difficulty and consensus distance in theory, while only empirically justify its generalization on training performance. As also pointed out by reviewers 1 and 3, the gap between the convergence rate/consensus distance and the generalization capability still exists, causing the mismatch between the theory and the simulations. But at the same time, the work can also serve as an initial good start and raises good points for the literature. I will keep my score unchanged.

---

> ### Author Response · Authors · 2020-11-19
> **Response to R4**
>
> We thank the reviewer for the time and valuable feedback. We will add corrections/clarifications as suggested. Please find answers to specific comments below:
>
> ### Convergence analysis v.s. Generalization performance
> Please refer to our [general response](https://openreview.net/forum?id=bIrL42I_NF8&noteId=R4dEk-WDEZM).
>
> ### The extension to the primal-dual methods
> Our paper aims to understand the limitations of decentralized SGD in deep learning for better algorithm design.
> To the best of our knowledge, we are not aware of practical primal-dual algorithms designed for decentralized deep learning. Extending primal-dual type algorithms to decentralized deep learning itself is non-trivial and is beyond the scope of this paper.
>
> We will add a discussion of these two papers in our related work section.

---

### Official Review · AnonReviewer1 · 2020-10-29
**An initial step towards understanding decentralized training**

**Rating:** 7
**Confidence:** 4

**Review:**

Summary:

This paper studies the problem of decentralized training where several computing units are used simultaneously to process the data, and computing units are assumed to be connected over a network. The main focus is to better understand the role of consensus, or lack there of, into the generalization abilities of decentralized training. The authors describe an upper bound for dissimilarity of local variables that guarantees the performance of decentralized training is as good as centralized one. Moreover, some heuristic guidelines are proposed to control consensus during training process. Some numerical evidence is also provided.

Reasons for score:

I believe the paper is well written and the results are useful for the literature. There are a couple of issues that need to be addressed. Moreover some context item that need to be elaborated more carefully.

Some items that need to be elaborated.

1. The main argument of the paper sees to be that generalization might be affected by decentralized training, as initially pointed out by Table 1.  However, at some point there is a conceptual leap and the discussion transforms into analyzing convergence rates. Although there is a connection between rates and generalization, one is not equivalent to the other. The provided analysis is done on rates, I do not think one can translate that into generalization so straightforward.

2. The authors claim to analyze the problem theoretically. However, wha seems to be the main result is left as Remark 2. I believe  Remark 2 is a statement that needs to proven, as represents the main issues addressed in the paper, namely, how consensus affects convergence rates.

3. The authors mention that the analysis is made on non-momentum algorithm, but experiments are made with the momentum version. This is an issue, as the translation of the obtained results into the momentum method needs to be proven. How are the authors sure that momentum does not play a role into the dependency on consensus?

4.  One main concept seems to be that \phi_t does not change too fast. This is left for the appendix. Such a main concept needs to be spelled out in the main text.

5. Is there a cite for Lemma 4?,  there seems to be studied in the literature before.

6. I read Remark 2 as the main result rather than Proposition 3.

7. I value the experimental results,  they are rather informative and complete.

---

> ### Author Response · Authors · 2020-11-19
> **Response to R1**
>
> We thank the reviewer for the time and valuable feedback. We will add corrections/clarifications as suggested. Please find answers to specific comments below:
>
> ### Convergence analysis v.s. Generalization performance (Question 1)
> Please refer to our [general response](https://openreview.net/forum?id=bIrL42I_NF8&noteId=R4dEk-WDEZM).
>
> ### Results on SGD without momentum (Question 3)
> Please refer to our [general response](https://openreview.net/forum?id=bIrL42I_NF8&noteId=R4dEk-WDEZM).
>
> ### Detailed derivatives for Remark2 (Question 2)
> We polished and re-formulated the derivatives for Remark 2 in Appendix A.1
>
> ### Answer to Question 6
> Thanks for your comment, we will rename Remark 2 as Proposition 2 as both are important.
>
> ### Reference for Lemma 4 (Question 5)
> We will add references in the new version.

---

### Official Review · AnonReviewer3 · 2020-11-01
**Comments on the effect of consensus in decentralized deep learning**

**Rating:** 4
**Confidence:** 4

**Review:**

This work investigated a very interesting topic about generalization in decentralized deep learning. The authors identify the consensus distance as the key factor that affects the generalization performance of decentralized training. In general, the paper is well written and there are several interesting observations and discoveries involved regarding the generalization performance of decentralized learning. But the quality and significance of the work seem not very high.

1. There is no clear link between the theory part and the numerical results. (Th1 is based on previous work.) The other results, e.g., remark 2, proposition 3, and lemma 4 cannot claim how the consensus distance affects the generalization error. All the statements are based on the observations in terms of consensus distance shown in eq4. I can only agree that the distance is related to the generalization error.

2. Also, Th1 quantified the convergence rate for SGD, while in the numerical results the authors used accelerated SGD and adam.

3. How did the critical distance be calculated? For example, what are L and sigma approximated?

4. From the numerical results, the authors at least claim two points of linking the critical consensus distance and the performance: i) the critical distance is important to the initial training phase; ii) a non-negligible consensus distance can improve the generalization performance. There is no convincing explanation that the critical distance contributes to the generalization. Also, there is no clear definition of either the initial training phase or middle phase, since the learning rate is chosen by the authors so that it might not reflect the true convergence phases. (especially in this case a warm-up scheme was used). In all, the discussion about the relation between the general error and critical distance is vague.

5. Except for the ring case, the generalization error of the most decentralized learning results might be worse than the centralization learning within 0.5%, which seems not that significant. Comparing the linear speed up benefited from the decentralized training, is this loss significant?

In summary, I don’t think the theory part is very strong in this paper, and the relation between the critical distance and the generalization error needs to be further justified.

---

> ### Author Response · Authors · 2020-11-19
> **Response to R3**
>
> We thank the reviewer for the time and valuable feedback. We will add corrections/clarifications as suggested. Please find answers to specific comments below:
>
> ### Convergence analysis v.s. Generalization performance (Question 1)
> Please refer to our [general response](https://openreview.net/forum?id=bIrL42I_NF8&noteId=R4dEk-WDEZM).
>
> ### Answer to the question "there is no convincing explanation that the critical distance contributes to the generalization" (Question 4)
> Please first refer to our *general response* titled “Convergence analysis v.s. Generalization performance”, where we explain how the critical distance derived from the convergence analysis can help us better understand the generalization. In addition to the general response, we also include some extra explanations below.
>
> Our claim for the critical distance for generalization is based on the extensive numerical results. For instance, in the case of node n=32 and dec-phase-1 in Table 2, generalization performance without consensus distance control is significantly lower than the fully centralized training upper bound (91.78 v.s. 92.82). However, one can recover the upper-bound performance (0 consensus distance) by reducing the distance to $1 / 4$ of the maximum distance in the case of without control, which satisfies our description of ‘critical distance’. *It demonstrates how the critical distance impacts the optimization quality of the critical initial training phase and thus impacts the final generalization*. More elaborate results can be found in Appendix Table 8.
>
> ### The definition of the ‘training phases’ and the choices of learning rate (Question 4)
> We follow the SOTA learning rate schemes for CV tasks in distributed deep learning [1, 2], thus we warm up the learning rate (over a very small fraction of training epochs: 5/300) and use stage-wise learning rates: our considered training phases are separated by the learning rate decay (i.e., epoch 0-150 and epoch 150-225, and epoch 225-300).
>
> Under this definition of the training phases, there is a dramatic difference between phases and high consistency within the phase. More precisely, within each phase, key properties related to optimization, such as gradient norm and smoothness (see Figure 5 in appendix), exhibit high consistency; besides, as shown in Figure 1, the consensus distance for normal decentralized training stays on the same level.
>
> The choice of the initial learning rate does not impact the division of the training phases: the value difference in the reasonable candidates of the learning rate is far less than the difference introduced by the learning rate decay (with the factor of 10).
> We do have similar observations even we use a different learning rate (c.f. Table 11 in appendix).
>
> ### Answer to the question “how was the critical distance calculated, how were L and sigma estimated?” (Question 3)
> We only empirically examined the existence of the *critical* consensus distance, and we did not compute the *critical* consensus distance in a closed-form.
> More precisely, we empirically measured and controlled the *consensus distance* as defined in Section 3.2 ($\Xi_{t}^{2} := \frac{1}{n} \sum_{i} || x_{i}^{(t)} - \bar{x}^{(t)} ||^2$), by controlling which we examined the existence of the *critical* consensus distance (as shown e.g. in Figure 2, Table 2 & 3 & 4 & 5).
>
> ### The significance of the drop in generalization (Question 5)
> Decentralized learning still encounters several quality drop issues on other communication topologies. We believe the mentioned 0.5% quality drop (e.g. from 92.82 to 92.27) on a relatively small scale graph (n=32) is significant for deep learning training. The gap will widen dramatically when considering the larger scale decentralized learning (e.g. the case of n=64 we considered in the main text).
>
> ### Results on SGD without momentum (Question 2)
> Please refer to our [general response](https://openreview.net/forum?id=bIrL42I_NF8&noteId=R4dEk-WDEZM).
>
> ### References
> 1. Accurate, Large Minibatch SGD: Training ImageNet in 1 Hour. Goyal et al, 2017.
> 2. Bag of Tricks for Image Classification with Convolutional Neural Networks. He et al, CVPR 2018.

---

### Author Response · Authors · 2020-11-19
**General response to all reviewers**

### Convergence analysis v.s. Generalization performance
Our paper first identifies the critical parameter to address the optimization difficulty in decentralized optimization (when comparing the convergence rate with that of centralized SGD). We theoretically derive the critical consensus distance based on the convergence analysis and empirically justify its impact/usefulness on training performance (as shown in Figure 2a & 2b) for decentralized deep learning. We further thoroughly examine the effectiveness of the proposed metric on the generalization performance (the main metric in deep learning).

**From convergence analysis to better understand generalization**. A line of recent research reveals the interference between initial training (optimization) [2, 3, 4] and the later reached local minima (generalization) [1, 5, 6, 7, 8]: the generalization of the deep nets cannot be studied alone via vacuous generalization bounds, and its practical performance is contingent on the critical initial learning (optimization) phase, which to some extent can be characterized by the conventional convergence analysis [2, 3, 4, 5, 6, 7].
This motivates us to derive the metric (i.e. critical consensus distance) from the convergence analysis, for the examination of the consensus distance (on different phases) in decentralized deep learning training. For example, (1) we identify the impact of different consensus distances at the critical learning phase on the quality of initial optimization, and the final generalization [2, 3, 4, 5] (i.e. our studied case of dec-phase-1), and (2) we reveal similar observations as in [5, 6, 7] when the optimization is no longer a problem (our studied case of dec-phase-2), where the existence of consensus distance can act as a form of noise injection [5] or sampling models from the posterior distribution [6, 7] (the detailed discussion can be found in our response to R2).

We will clarify these points in the next revision.

### Results on SGD w/ and w/o momentum
Current analysis on SGD with momentum is rather loose and does not characterize the acceleration benefit observed in deep learning practice. Thus, building the consensus distance analysis on top of loose momentum analysis may not be as meaningful.
Our work aims to better understand the limitations of SOTA training methods in decentralized deep learning and thus our empirical understandings are on top of these SOTA training schemes. We additionally include experiments on SGD without momentum, to demonstrate that our claim on the relation between consensus distance and generalization performance stands regardless of the use of momentum.

Numerical Results on Vanilla SGD (n = 32, 64). For n=32, 64, we use the scaling-up factor of 32, 64 respectively, and run experiments over 3, 2 seeds respectively. We can observe the consistent pattern as in the case of SGD with Nesterov momentum presented in our main paper. This validates the coherence between our theory and experiments. Note in the case of $n=32$, the performance gap between ‘ring’ and ‘complete’ is not significant, however, the pattern manifests in the case $n=64$ where the performance gap is considerable.

|             	|                 	| n=32             	| n=64              	|
|-------------	|-----------------	|------------------	|-------------------	|
| ring        	|                 	| $90.30 \pm 0.14 \quad$ 	| $88.92 \pm 0.23$  	|
| complete    	|                 	| $90.64 \pm 0.19$ 	| $90.58 \pm 0.26 $ 	|
| dec-phase-1 	| $\quad \Xi_{max} \quad$     	| $90.51 \pm 0.05$ 	| $88.80 \pm 0.03$  	|
|             	| $\quad 1/2 \Xi_{max} \quad$ 	| $90.74 \pm 0.14$ 	| $89.89 \pm 0.03$  	|
|             	| $\quad 1/4 \Xi_{max} \quad$ 	| $90.88 \pm 0.37$ 	| $90.43 \pm 0.05$  	|
| dec-phase-2 	| $\quad \Xi_{max} \quad$     	| $90.64 \pm 0.18$ 	| $90.63 \pm 0.37$  	|
|             	| $\quad 1/2 \Xi_{max} \quad$ 	| $90.55 \pm 0.19$ 	| $90.46 \pm 0.15$  	|
|             	| $\quad 1/4 \Xi_{max} \quad$ 	| $90.57 \pm 0.17$ 	| $90.63 \pm 0.25$  	|

### Reference
1. Implicit Regularization in Deep Learning. Behnam Neyshabur, Ph.D. Thesis, 2017.
2. The Break-Even Point on the Optimization Trajectories of Deep Neural Networks. Jastrzebski et al, ICLR 2020.
3. Time Matters in Regularizing Deep Networks: Weight Decay and Data Augmentation Affect Early Learning Dynamics, Matter Little Near Convergence. Golatkar et al, NeurIPS 2019.
4. Critical Learning Periods in Deep Networks. Achille et al, ICLR 2019.
5. Don't Use Large Mini-Batches, Use Local SGD. Lin et al, ICLR 2020.
6. Stochastic Weight Averaging in Parallel: Large-Batch Training that Generalizes Well. Gupta et al, ICLR 2020.
7. Averaging Weights Leads to Wider Optima and Better Generalization. Izmailov et al, UAI 2018.
8. On Large-Batch Training for Deep Learning: Generalization Gap and Sharp Minima. Keskar et al, ICLR 2017.

---

### Decision · Program_Chairs · 2021-01-07
**Final Decision**

**Decision:**

Reject

**Comment:**

The authors study the problem of (insufficient) generalization in gossip-type decentralized deep learning. Specifically, they establish an upper bound on the square of the consensus parameter distance, which the authors identify as a key quantity that influences both optimization and generalization. This upper bound (called the critical consensus distance) can be monitored and controlled during the training process via (e.g.) learning rate scheduling and tweaking the amount of gossip. A series of empirical results on decentralized image classification and neural machine translation are presented in support of this observation.

Initial reviews were mixed. While all reviewers liked the approach, concerns were raised about the novelty of the results, the lack of theoretical depth, and the mismatch between theory and experiments. Overall, the idea of tracking consensus distance to control generalization seems to be a practically useful concept.  During the discussion phase the authors were been able to (convincingly, in the area chair's view) respond to a subset of the criticisms.

Unfortunately, concerns remained regarding the mismatch between the theoretical and empirical results, and in the end the paper fell just short of making the cut.

The authors are encouraged to carefully consider the reviewers' concerns while preparing a future revision.